# A Novel Approach for High-Performance Estimation of SPI Data in Drought Prediction

Levent Latifoğlu [1],* and Mehmet Özger [2]

1   Civil Engineering Department, Erciyes University, Kayseri 38039, Türkiye
2   Civil Engineering Department, Istanbul Technical University, Istanbul 34469, Türkiye; ozgerme@itu.edu.tr
*   Correspondence: latifoglu@erciyes.edu.tr

**Abstract:** Drought, as a natural disaster, has significant negative consequences and directly impacts living organisms. Drought forecasting commonly relies on various drought indices, with the Standardized Precipitation Index (SPI) being widely used. In this study, we propose a novel approach to estimate SPI values at 3- and 6-month lead times with high accuracy. This novel method introduces a phase transfer entropy (pTE) technique that analyzes time-shifted data matrices and the connectivity of SPI-3 and SPI-6 data. By maximizing the information flow between these data points, the most suitable time index $(t - n)$ for input data in forecasting models is determined. This approach, not previously explored in the literature, forms the basis for predicting SPI values effectively. Machine learning algorithms, in combination with the Tunable Q Factor Wavelet Transform (TQWT) optimized by the Grey Wolf Optimization (GWO) algorithm, are employed to predict SPI values using the identified input data. The TQWT method generates subband signals, which are then estimated using Artificial Neural Networks (ANN), Support Vector Regression (SVR), and the Gaussian Process Regression Model (GPR). To evaluate the performance of the proposed GWO-TQWT-ML models, the subband data derived from the SPI is also estimated using ANN, GPR, and SVR models with the Empirical Mode Decomposition and Variational Mode Decomposition methods. Additionally, non-preprocessed SPI data is estimated independently using ANN, GPR, and SVR models. The results demonstrate the superior performance of the pTE-GWO-TQWT-ML models over other methods. Among these models, the pTE-GWO-TQWT-GPR model stands out with the best prediction performance, surpassing both the pTE-GWO-TQWT-ANN and pTE-GWO-TQWT-SVR models. The pTE-GWO-TQWT-GPR model yielded determination coefficient $(R^2)$ values for SPI-6 data as follows: 0.8039 for one-input, 0.9987 for two-input, and 0.9998 for three-input one ahead prediction, respectively; 0.9907 for two-input two ahead prediction; and 0.9722 for two-input three ahead prediction. For SPI-3 data, using the pTE-GWO-TQWT-GPR model, the $R^2$ values were as follows: 0.6805 for one-input, 0.9982 for two-input, 0.9996 for three-input one ahead prediction, 0.9843 for two-input two ahead prediction, 0.9535 for two-input three ahead prediction, 0.9963 for three-input two ahead prediction, and 0.9826 for three-input three ahead prediction. Overall, this study presents a robust method, the pTE-GWO-TOWT-GPR model, for the time series estimation of SPI data, enabling high-performance drought prediction.

**Keywords:** drought; forecasting; Standardized Precipitation Index; SPI; phase transfer entropy; Grey Wolf Optimization; Tunable Q Factor Wavelet Transform; machine learning

## 1. Introduction

Drought is a natural climatic event that significantly affects living things and can bring about serious problems. As a result of the precipitation falling significantly below the recorded normal levels, it causes the land and water resources to be adversely affected and the hydrological balance to deteriorate. Many problems arise as a result of drought, such as a decrease in water quantity and quality, land and soil degradation, decrease in agricultural productivity, and desertification [1–3].

Drought is divided into four subtitles as meteorological, agricultural, hydrological, and socioeconomic [4]. The decrease in precipitation below the average for a certain period is defined as meteorological drought, and it is the difference between the annual, seasonal, or monthly precipitation totals from the average [5]. Agricultural drought, which expresses the lack of water in the soil to meet the needs of the plant, causes a slowdown in plant growth, loss of crops, and is an important threat to animals [6]. As a result of the hydrological drought that develops after a prolonged meteorological drought, groundwater, springs, runoff, and soil moisture are affected, and sharp decreases are observed in lakes, rivers, and groundwater [7]. Drought begins as a meteorological drought, develops as an agricultural and hydrological drought, and its effects become visible as a socioeconomic drought. Unlike natural disasters, drought severely affects a wide variety of lands, agriculture, and socioeconomic infrastructure and is a far more complex natural hazard [8]. The intensity and characteristics of drought vary from region to region, and many regions of the world are in danger of drought. Due to drought, more than 650,000 deaths were reported by the World Meteorological Organization (WMO) from 1970 to 2019 [3,9]. In recent years, the issue of drought has been addressed in studies on climate change, and many studies have been carried out to analyze and predict drought risk. For this reason, many modeling approaches are being developed that give an idea about the amount of the precipitation and ultimately improve the ability to monitor droughts. Drought risk analysis aims to improve drought management and forecasting techniques, and drought risk analysis focuses on the size, duration, intensity, and spatial extent of droughts, taking into account the spatial variability of the drought [10]. The drought formation time is slow, and the consequences of a drought appear over a long period of time relative to its onset and the time it is perceived by ecosystems and hydrological systems. For this reason, a forecasting system that can warn about a drought at the beginning will be able to significantly reduce the negative effects of the drought [11]. While all types of the droughts begin with a lack of precipitation in time and/or space, an early stage of a lack of precipitation accumulation often manifests as a meteorological drought. In order to monitor and predict droughts, various existing drought indices are used to determine the deviation of meteorological variables, such as precipitation, from their long-term averages [12]. Drought monitoring relies on various indices, including the Standardized Precipitation Evapotranspiration Index (SPEI) [13], Palmer Drought Severity Index (PDSI) [14], Effective Drought Index (EDI) [15], Reconnaissance Drought Index (RDI) [16], Keetch–Byram Drought Index (KBDI) [17], and Weighted Anomaly Standardized Precipitation Index (WASP), as well as Standardized Drought Indices (SDIs) [18,19] A comprehensive list of these indices and their descriptions is available in Reference [20]. However, the most widely used method for drought monitoring globally is the SDI-type procedure [19]. Some recently developed SDI-type drought indices include the Standardized Precipitation Index (SPI) [21].

The Standardized Precipitation Index (SPI), developed by McKee et al., is used as a global drought monitoring tool following the Lincoln Drought Declaration of the World Meteorological Organization [21–23]. SPI calculations are easier than other indices, because it only requires precipitation data. Thus, it is very useful in drought risk analysis and estimations in regions where data are scarce and where other parameters, such as stream flow, evapotranspiration, and soil moisture information, may not be readily available. The SPI is comparable in both time and space, and also, it can be calculated for multiple time scales [24,25]. Thus, it allows to determine the duration, size, and intensity of droughts and defines various drought types as hydrological, agricultural, or environmental. Its calculation based on a probabilistic basis has led to an important place in drought analysis and is widely used for drought analysis in many regions of the world. Therefore, the SPI estimation is important in drought analysis.

To date, numerous research papers have been published to estimate drought indices using data-driven techniques. In the development of data-based drought forecasting models from these studies, linear approaches, in which recursive moving average (ARMA) models

are used, have been replaced by nonlinear approaches based on Artificial Intelligence models in recent years [26,27].

In a previous study by Mokhtarzad et al., drought prediction was conducted using the Standardized Precipitation Index (SPI) with three different modeling techniques: Artificial Neural Network (ANN), Support Vector Machine (SVM), and Adaptive Neuro-Fuzzy Interface System (ANFIS). Notably, the SVM model outperformed both the ANN and ANFIS models in terms of predictive performance [26]. Pande et al. evaluated various machine learning (ML) models, including Artificial Neural Network (ANN) and M5P Tree, for predicting the Standardized Precipitation Index (SPI) at different time scales (SPI-3 and SPI-6). The research utilized rainfall data from two Indian stations, Angangaon and Dahalewadi. Among the models tested, the M5P model outperformed the others, particularly in terms of r values and RMSE values, making it the superior choice for drought prediction at both stations [28]. In addition, in a literature study, a SPI estimation study was performed on a time scale of 1–12 months using different Adaptive Neuro-Fuzzy Inference System (ANFIS) models, and it was shown to be a suitable model for SPI estimations [29]. In another study performed for the estimation of the drought index, a wavelet-based drought model using the extreme learning machine (W-ELM) algorithm was used, and it was seen that the wavelet transform increased the prediction performance in drought prediction [30]. In a study showing that subband decomposition methods increase the estimation performance in SPI estimations with the Variational Mode Decomposition (VMD), Discrete Wavelet Transform (DWT), and Empirical Mode Decomposition (EMD) methods. In this study, it was observed that the VMD-GPR model demonstrated a better forecasting performance for the 1- and 3-month time scale SPI, while the DWT-ANN model exhibited superior performance in forecasting the 6-month time scale SPI data [31]. In another study, SPI data werewere estimated using wavelet packet-genetic programming, the performance of this model was compared with the Autoregressive (AR1), GP, and Random Forest (RF) models, and the forecasting success of wavelet packet-genetic programming was demonstrated [32]. As evident from the literature, hybrid approaches that estimate the SPI from subband signals by decomposing the data using preprocessing methods have shown an enhanced forecasting performance. Nevertheless, it is crucial to note that boundary effects pose significant challenges, particularly in methods like EMD and WT [33,34]. In addition, it is seen that the estimation performance changes depending on the wavelet function used in the wavelet transform [31].

The VMD method employs spectral segmentation of the Fourier spectrum for signal decomposition into subbands. Consequently, it shares a common constraint with the Fourier spectrum [35]. TQWT, on the other hand, is a method that, unlike conventional signal preprocessing techniques based on frequency bands such as WT and EMD, can decompose the targeted signal into specific components with different structures according to their oscillation characteristics and eliminate in-band noise [35].

In this study, a novel forecasting framework has been introduced for SPI data on 3- and 6-month time scales. The notation SPI-3 is used for the 3-month time scale, while SPI-6 is used for the 6-month time scale.

One of the unique contributions of this study is the utilization of a connectivity analysis using the pTE approach for SPI data prediction, specifically focusing on determining the optimal time lag values and SPI values at these time indices for input in the forecasting model. This novel approach represents the first time in the literature for identifying the most suitable input SPI-3 and SPI-6 data with the optimal time indices for the prediction model. In this method, the PTE values were calculated using a 1–3 phase delay from the Hankelization matrix obtained through time lag operations, thereby enabling the identification of time lags where the information flow is at its maximum and minimum. By concentrating on time lags with maximum information flow, a methodology has been established for determining the most appropriate time indices and SPI inputs for use as input in SPI prediction. Additionally, the TQWT method was employed to decompose SPI-3 and SPI-6 data into subbands, and the subband signals were subsequently estimated

using ANN, SVR, and GPR models. The optimization of the TQWT method involved adjusting parameters like the Q Factor, subband decomposition level, and the parameters of the machine learning models. This optimization was achieved through the GWO algorithm, resulting in the introduction of the GWO-TQWT-ML method, a novel approach in the literature known for its high forecasting performance with SPI data. To evaluate the performance of the GWO-TQWT-ML method, we compared it against various models, including EMD-ANN, EMD-SVR, EMD-GPR, VMD-ANN, VMD-SVR, VMD-GPR, ANN, SVR, and GPR models, with the aim of benchmarking its performance. The study's results were meticulously assessed using performance metrics such as MSE, MAE, R, $R^2$, and scatter plots. In summary, this study represents a significant advancement in SPI drought index estimations by introducing the novel pTE-GWO-TQWT-GPR approach. This approach leverages the maximum information flow to determine input features for the forecasting model, and its incorporation significantly enhances the forecasting accuracy.

## 2. Materials and Methods

The operational steps carried out in this study are presented in the provided flowchart Figure 1.

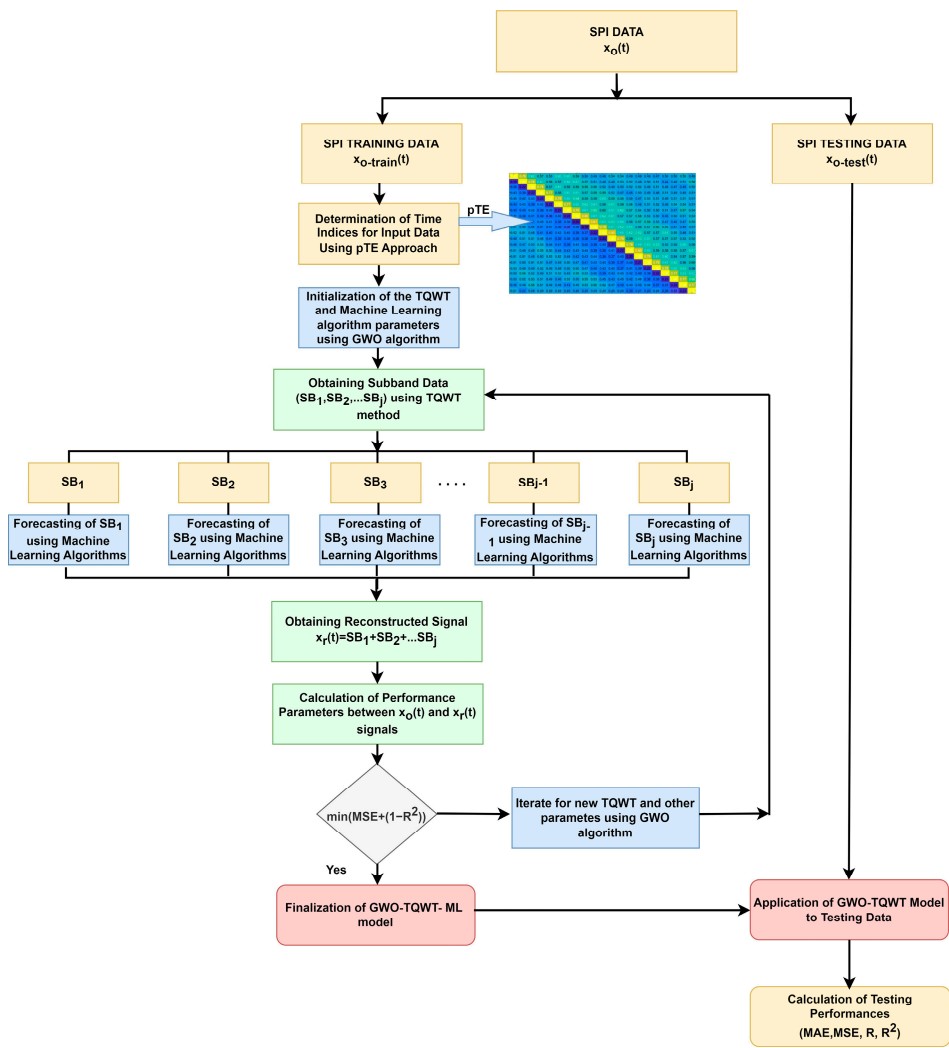

**Figure 1.** Flowchart of the proposed SPI forecasting model.

### 2.1. Study Area and Data

In this study, precipitation data were utilized from the Catchment Attributes and Meteorology for Large-sample Studies (CAMELS) dataset [36–38]. The CAMELS dataset

comprises observed runoff levels and catchment-aggregated meteorological forcing data at the daily time scale. Daily meteorological data were computed using gridded data sources, including the Daymet data [39], the Maurer data [40], and the National Land Data Assimilation System (NLDAS) data [41] in the CAMELS dataset. Interested readers can find comprehensive information about this dataset [36–38].

For the proposed drought forecasting study, the Daymet data, due to its high spatial resolution, is crucial for accurately estimating spatial variability in basins with complex topography. You can access these parameters freely at https://ral.ucar.edu/solutions/products/camels (accessed on 1 February 2022) [36,37]. In this study, the SPI-3 and SPI-6 indices were computed using precipitation data collected from the gauge with ID 13340000 from the CAMELS dataset, spanning from 1 October 1980 to 31 December 2014. This gauge is situated in the Clearwater River Basin in Orofino, characterized by a substantial drainage area of 14,268.92 km$^2$ and geographic coordinates at latitude 46°28′42″, longitude 116°15′27″. The study area is seen in Figure 2.

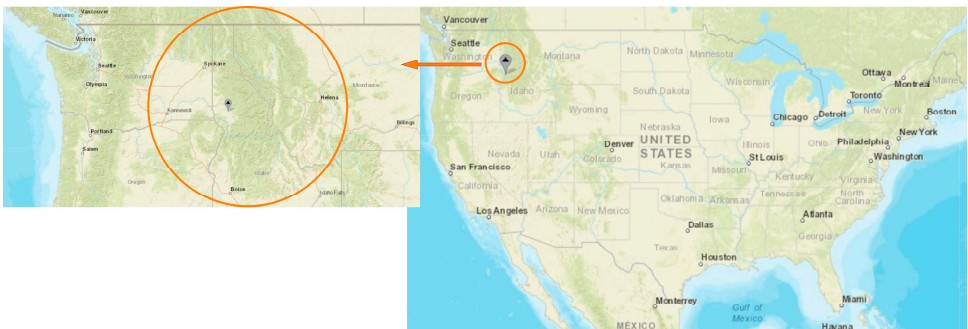

**Figure 2.** Study area [42].

The precipitation data in this database are daily data, and monthly averages of these data are taken, as seen in Figure 3.

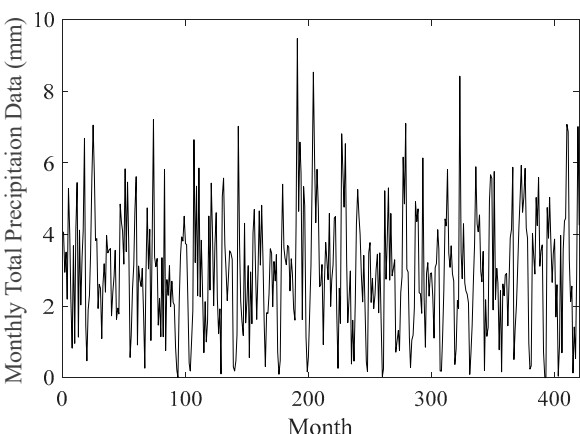

**Figure 3.** Precipitation data.

Standardized Precipitation Index

The SPI is a drought index that only uses precipitation as a variable. As a versatile tool for drought monitoring and analysis, the SPI is already widely employed on a global scale [21]. Anywhere in the world, dry or humid conditions and anomalies may be identified on a certain time scale using SPI and precipitation data records. Different time intervals, such as monthly and annual, can be used to monitor the drought situation.

The Standardized Precipitation Index (SPI) calculation involves several key steps. First, a Probability Density Function (PDF) is applied to the precipitation data. The choice of the parametric distribution at this stage significantly affects the SPI's precision.

McKee et al. [21] recommended the use of the gamma distribution, especially for precipitation records covering extensive time periods. Gamma distribution is described by Equation (1).

$$(x) = \frac{1}{\beta^\alpha \tau(\alpha)} Y^{\alpha-1} e^{-Y/\beta} \tag{1}$$

where $\alpha > 0$ shows the shape parameter, $\beta > 0$ is the scale parameter, $\tau$ is the gamma function and $Y > 0$ is the precipitation amount.

Next, a cumulative probability density distribution is generated for the specific time scale under consideration. For time periods exceeding six months, the central limit theorem suggests that the observed probability distributions tend to become standardized. Consequently, it becomes feasible to substitute the gamma distribution with a normal probability distribution, which can simplify computations and enhance the accuracy. To transform the cumulative probability distribution into a standard normal distribution with a mean of zero and a variance of one, the following equation (Equation (2)) is applied.

$$SPI = \frac{Y_i - \overline{Y_i}}{\sigma_i} \tag{2}$$

SPI offers a comprehensive view of the impact of precipitation scarcity on water resources, with the drought severity indicated by negative SPI values. For detailed drought categorization, refer to Table 1.

**Table 1.** Classification of the SPI.

| SPI Threshold | Classification |
|---|---|
| SPI $\geq$ 2 | Extreme Wet |
| $1.50 \leq$ SPI $< 2$ | Very Wet |
| $1.0 \leq$ SPI $< 1.5$ | Moderate Wet |
| $-1.0 \leq$ SPI $< 1.0$ | Near normal |
| $-1.5 \leq$ SPI $< -1.0$ | Moderate Drought |
| $-2.0 \leq$ SPI $< -1.5$ | Severe Drought |
| SPI $< -2.0$ | Extreme Drought |

In this study, we focused on SPI-3, which represents precipitation conditions over a 3-month period, and SPI-6, which represents precipitation conditions over a 6-month period. These SPI-3 and SPI-6 data shown in Figure 4 were calculated using DRIN-C programs from monthly precipitation data [43,44]. In the analysis, 70% of the SPI data were allocated for training, while the remaining 30% was reserved for testing, without any randomization or reordering. This separation allowed us to evaluate the forecasting model's performance effectively.

The selection of the ideal number of delays (m) and input variables, which have a significant impact on the precision and complexity of the predictions, is a crucial stage in drought forecasting. An insufficient number of delays or excessive number of delays can result in poor or complex solutions, respectively. In general, the number of delays has been defined by the trial and error method or by the analysis of the autocorrelation function (ACF) of the drought index. However, the ACF analysis is a linear approach, and the results are frequently extremely big; because of this, the process is more complex and takes more time.

In contrast, the novelty of this study lies in the application of the pTE approach, which was used for the first time in the literature to determine input variables and optimal time delay for forecasting purposes. pTE provides a robust and effective tool for identifying the optimal time delay and input variables for the proposed drought forecasting model, allowing us to discern how one event influences another. Traditional methods often cause a time-consuming and complex analysis for forecasting. However, pTE streamlines this process significantly while yielding more precise and efficient results. Thus, by using the pTE approach to determine the optimal time delays and input variables, it significantly

contributes to improving the forecasting performance and enables the development of prediction models in a more effective manner.

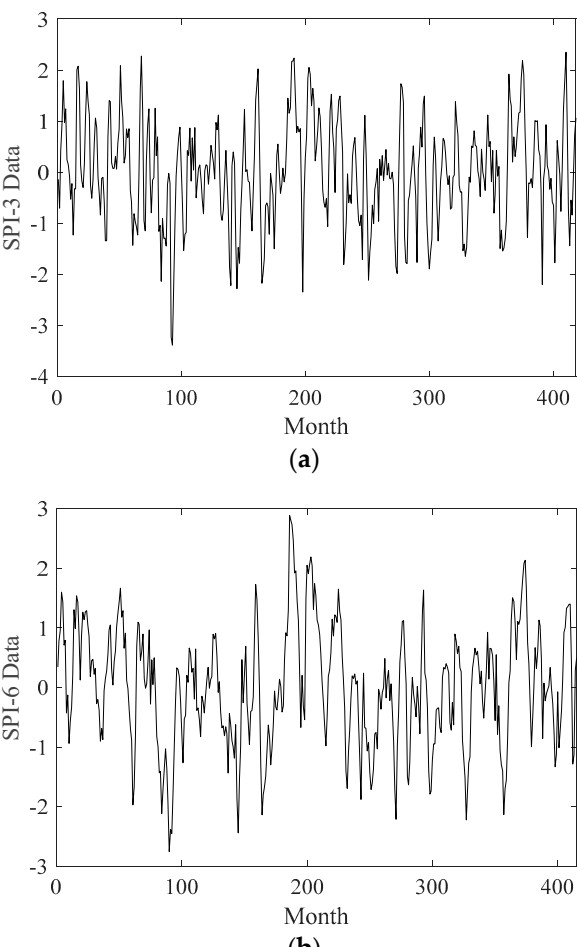

**Figure 4.** (**a**) SPI-3 and (**b**) SPI-6 data.

### 2.2. Phase Transfer Entropy

Transfer entropy (TE) [45] was introduced by reformulating the Wiener principle [46] within the framework of information theory (IT) [47]. The TE estimates whether the inclusion of the history of both the source and target time series affects its ability to predict the future of the target time series. Transfer entropy (TE) compares the conditional probabilities using Kullback–Leibler divergence and states that the probability density of the future of Y conditioned on its own past should differ from the probability density of the future of Y conditioned on the pasts of both X and Y if the signal X causes the signal Y.

To measure phase-to-phase interactions between causal or directional domains, a phase-based measure of effective coupling is needed. There are two approaches to detecting effective connectivity: model-based (e.g., Dynamic Causal Modeling [48]) and model-free techniques (e.g., Granger Causality [49]). Transfer entropy (TE) [45] is a model-independent measure of effective coupling based on information theory. However, previous applications of TE have dealt with the connectivity analysis of a real-valued time series where the signal amplitude is a large deviation and the phase is only an implicit variable. Also, the current TE methods have limited robustness against narrowband filtering, noise, and linear mixing between signals. Finally, the approaches used for TE depend on an accurate a priori estimate of some parameters. These situations create a disadvantage for TE. Therefore, a new measure, phase transfer entropy (Phase TE), has been introduced to predict the directional correlation between the complex phase time series of the signal. Phase transfer entropy (PTE), introduced by Lobier et al. [50], is used as a measure of the strength and

direction of the information flow between signals. Phase TE is computationally effective and resistant to noise and linear mixing, which are crucial factors in time series data. Considering that pTE is also essentially parameter-dependent, it can be used as an efficient and reliable method to evaluate effective connectivity in time series signals.

If the instantaneous phase of a x(t) time series signal is expressed as θ(t), the phase transfer entropy is expressed by Equation (3).

$$
\begin{aligned}
Phase\ TE_{X \rightarrow Y} = \\
H(\theta_y(t), \theta_y(t')) + H(\theta_y(t'), \theta_x(t')) - H(\theta_y(t')) - H(\theta_y(t), \theta_y(t'), \theta_x(t'))
\end{aligned}
\tag{3}
$$

where $\theta_y(t')$ and $\theta_x(t')$ are the values of the time point t at the time $\varsigma$ and time lag of t.

The marginal entropy and joint entropy expressions are seen in Equations (4)–(7), and detailed information on the phase TE is given in Ref [50].

$$
H(\theta_y(t), \theta_y(t')) = -\sum p(\theta_y(t), \theta_y(t')) \log H(\theta_y(t), \theta_y(t'))
\tag{4}
$$

$$
H(\theta_y(t'), \theta_x(t')) = -\sum p(\theta_y(t'), \theta_x(t')) \log H(\theta_y(t'), \theta_x(t'))
\tag{5}
$$

$$
H(\theta_y(t')) = -\sum p(\theta_y(t')) \log H(\theta_y(t'))
\tag{6}
$$

$$
\begin{aligned}
H(\theta_y(t), \theta_y(t'), \theta_x(t')) = \\
-\sum p(\theta_y(t), \theta_y(t'), \theta_x(t')) \log H(\theta_y(t), \theta_y(t'), \theta_x(t'))
\end{aligned}
\tag{7}
$$

Application of Phase Transfer Entropy to SPI Data

In this study, a time-shifted data matrix was created by shifting the SP-3 and SPI-6 training data. The obtained data matrix is seen in Equation (8).

$$
\begin{bmatrix}
SPI(t) & SPI(t-1) & SPI(t-2) & \ldots & SPI(t-\mathbf{m}) \\
SPI(t+1) & SPI(t) & SPI(t-1) & \ldots & SPI(t-\mathbf{m}+1) \\
\vdots & \vdots & \vdots & \ldots & \vdots \\
SPI(t+\mathbf{n}) & SPI(t+\mathbf{n}-1) & SPI(t+\mathbf{n}-2) & \ldots & SPI(t+n-m)
\end{bmatrix}_{n x m}
\tag{8}
$$

In order to measure the information flow between the obtained delayed time series, pTE values (named TE1–TEm) were calculated separately for the 1, 2, and 3 phase values, and TE1-1, TE1-2, TE1-3, . . . TEm-1,TEm-2,TEm-3 data were obtained. The pTE values, where the phase delay was taken as 1, were used in the one ahead forecasting model, the pTE values where the phase delay was taken as 2 were used in the two ahead forecasting model, and the pTE values where the phase delay was 3 were used in the three ahead forecasting model for feature selection. The flowchart of the application of phase transfer entropy is shown in Figure 5.

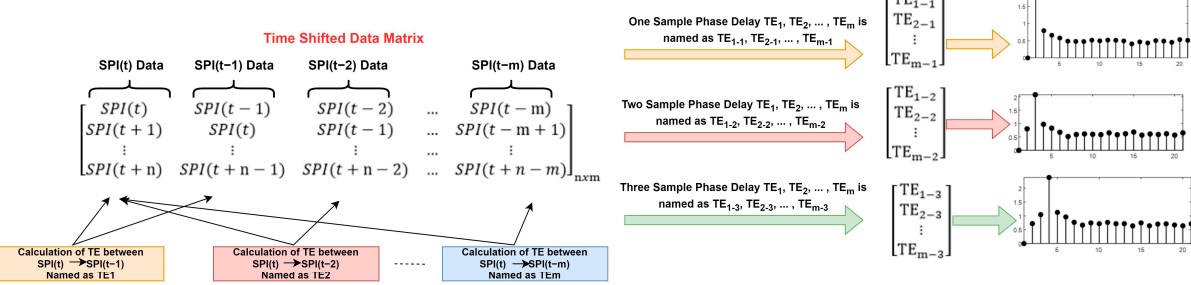

**Figure 5.** Flowchart for the pTE calculations.

### 2.3. Tunable Q Wavelet Transform

In this study, the TQWT method was employed for the efficient analysis of oscillating signals by adjusting parameters such as the Q Factor, redundancy factor (r), and the number of decomposition levels (J) [51]. TQWT allows for multilevel signal decomposition by applying low-pass and high-pass filter banks, generating high-pass subband (HPS) and low-pass subband (LPS) signals. These subbands are subsequently combined to reconstruct the input signal. The Q and r parameters influence the passband width of each filter, and TQWT offers good time domain localization of wavelets due to its oversampled filter bank characteristic [51,52]. The study focused on optimizing the TQWT parameters, specifically Q and J, using the Grey Wolf Optimization (GWO) algorithm, demonstrating the algorithm's effectiveness in determining these parameters for drought prediction in SPI data.

### 2.4. Grey Wolf Optimization Algorithm

The GWO algorithm, inspired by the hunting behavior of grey wolves, consists of four hierarchical groups of wolves: alpha, beta, delta, and omega. These wolves represent the leadership hierarchy, with alpha being the top leader. The algorithm simulates the hunting process, starting with locating prey under the alpha wolf's guidance, and alpha, beta, and delta wolves estimate the prey's location. Other search agents update their positions accordingly [53].

During the hunting phase, wolves can move randomly around their prey based on specific equations. When the prey stops moving, the wolves attack. The algorithm's core elements involve modifying coefficients A and B, which influence the algorithm's search behavior. Varying these coefficients allows the algorithm to explore both local and global solutions [53].

In the context of this study, the GWO algorithm is used to optimize the parameters in the Tunable Q Factor Wavelet Transform (TQWT) for drought prediction. It offers an effective means of parameter optimization for improved forecasting.

### 2.5. Artificial Neural Networks

ANNs are inspired by the human brain and are used for various applications, including drought prediction. ANNs consist of artificial neurons with inputs, weights, activation functions, and outputs. Inputs are the information received by the neuron, while weights represent the influence of the inputs on the neuron's output [54].

In this study, Multi-Layered Feed-Forward Artificial Neural Networks (MLP) are used, which consist of input, hidden, and output layers. The number of neurons in the hidden layer was optimized using the Grey Wolf Optimization (GWO) algorithm. The learning in ANNs involves adjusting weights with training algorithms. In this study, the Levenberg–Marquardt algorithm, known for its high performance, was used for MLP network training.

### 2.6. Support Vector Regression

SVM are powerful tools for classification and regression in machine learning. Support Vector Regression (SVR), an adaptation of SVM for regression tasks, aims to find a function that approximates the training data while minimizing the estimation errors. It is particularly useful for solving nonlinear problems [55].

SVR involves mapping input data to a higher-dimensional space using a nonlinear mapping function $\varphi$ and finding adjustable coefficients $w$ and $b$ to represent the relationship between inputs and outputs. The SVR optimization problem aims to minimize a cost function with constraints involving training errors $\xi_i$ and $\xi_i^*$. The parameter vector $w$ is determined through Lagrange multipliers, resulting in the SVR formula [56].

### 2.7. Gaussian Process Regression Model

GPR is a probabilistic model used for predicting output variables based on input data. It leverages probability theory to provide precise predictions and has gained popularity in forecasting studies. GPR offers flexibility through various covariance functions, allowing researchers to choose the most suitable one for their specific needs [57].

In GPR, the Gaussian process defines the distribution of functions, where $f(x)$ is characterized by its mean $m(x)$ and covariance $k(x,x')$. The relationship between the input and output vectors is expressed by $y_i = f(x_i) + \varepsilon$, with $\varepsilon$ representing normally distributed noise [58].

### 2.7.1. Empirical Mode Decomposition

The EMD method, developed by Huang et al., decomposes complex signals into Intrinsic Mode Functions (IMFs) and a residue. IMFs are relatively stationary subseries extracted based on the local characteristic scale of the signal. An IMF is defined by two conditions: the sum of extremes and zero crossings must be equal (or differ by one at most) across the entire dataset, and the mean of the envelope defined by the local maxima and minima must be zero at any point [57].

EMD involves iteratively extracting IMFs from the original time series, starting by defining extremums, calculating the upper and lower envelopes, and deriving the mean envelope. The mean envelope is then subtracted from the original signal to test if the result is an IMF. If it meets the criteria, it becomes an IMF, and the process continues with the residual signal. This iterative process continues until the residual satisfies the stopping conditions [58].

### 2.7.2. Variational Mode Decomposition

The VMD method is an adaptive non-recursive technique for decomposing signals into different modes. Each mode in VMD is represented as a band-limited signal characterized by its center frequency, instantaneous amplitude, and instantaneous frequency [59]. The method aims to decompose a signal into these modes by optimizing a constrained variational model. The key equations in VMD involve representing each mode as a cosine function with varying amplitude and frequency, estimating the bandwidth using Gaussian smoothness, and using optimization techniques like the Lagrangian function and the Alternate Direction Method of Multipliers (ADMM) to find the optimal solution for the constrained variational model. In the VMD algorithm, each mode is updated iteratively in the frequency domain, and the center frequency is recalculated at each step [59].

### 2.8. Model Evaluation

The performance of the proposed hybrid model for forecasting droughts is measured by considering the mean absolute error (MAE), mean square error (MSE), correlation coefficient (R), and determination coefficient ($R^2$) values [60]. The formulas of the used performance metrics are presented in Table 2.

**Table 2.** Performance evaluation parameters used in the study.

| Metric | Description | Formula |
|---|---|---|
| MAE | Mean Absolute Error | $MAE = \frac{1}{N} \sum\limits_{i=1}^{N} \left\| X_{obtained,i} - X_{forecasted,i} \right\|$ |
| MSE | Mean Square Error | $MSE = \frac{1}{N} \sum\limits_{i=1}^{N} \left( X_{obtained,i} - X_{forecasted,i} \right)^2$ |
| R | Correlation Coefficient | $R = \frac{1}{N-1} \sum\limits_{i=1}^{N} \left( \frac{X_{obtained,i} - \mu_X}{\sigma_X} \right) \left( \frac{X_{forecasted,i} - \mu_{Xtah}}{\sigma_{Xtah}} \right)$ |
| $R^2$ | Determination Coefficient | $R^2 = 1 - \frac{\sum_{i=1}^{N} \left[ X_{obtained,i} - X_{forecasted,i} \right]^2}{\sum_{i=1}^{N} \left[ X_{obtained,i} - \mu_X \right]^2}$ |

### 2.9. Framework of the Developed pTE-GWO-TQWT-ML Hybrid Model for Drought Forecasting

The parts of the novel combined modeling framework for drought forecasting in this study include a novel connectivity analysis and input determination using the pTE stage, a novel forecasting model created by combining the TQWT decomposition method and ML algorithm with the GWO method for optimizing the TQWT and ML parameters, and the stage of model evaluation.

#### 2.9.1. Connectivity Analysis and Input Variable Determination Phase

Selection of the input variables and the determination of the time lag are very important in forecasting studies. In this study, the pTE approach was used for the first time in the literature to analyze the information flow of shifted time series data and to determine its connectivity.

In this study, the normalized pTE values for SPI-3 and SPI-6 data were computed with a one-phase delay for the one-step ahead prediction model, with a two-phase delay for the two-step ahead prediction model, and with a three-phase delay for the three-step ahead prediction model from the Hankelization matrix. The obtained normalized pTE values were used as inputs for the GWO-TQWT-ML model, where, for the 1-input model, the SPI data corresponding to the time lag index at which the highest pTE value occured were used. For the 2-input model, the SPI data corresponding to the time lag indices where the highest two pTE values occured were utilized. Similarly, for the 3-input model, the SPI data corresponding to the time lag indices where the highest three pTE values occured were employed.

Furthermore, in this study, a maximum delay of 20 was chosen for the Hankelization process, with the aim of examining the connectivity and information flow between the SPI data up to 20 time delays. Here, the value of 20 is chosen as an example. If the connectivity between SPI data at different time delay numbers is to be investigated, the maximum time delay can be adjusted accordingly.

In Figure 6a, normalized pTE values calculated with a one-phase delay for one ahead forecasting are shown, demonstrating the connectivity and information flow between SPI $(t - \varsigma)$ and SPI $(t - \tau)$ data with time lag indices of $\varsigma$ $(t - \varsigma)$ and $\tau$ $(t - \tau)$ for both SPI-3 and SPI-6 data separately for $\varsigma = 0,1,2,\ldots,20$ and $\tau = 0,1,2,\ldots,20$.

As can be seen from Figure 6a, as an example, the normalized one-phase delay pTE value between the SPI-3 $(t - 1)$ data and the SPI-3 $(t)$ data $(pTE_{SPI3(t-1) \to SPI3(t)}$, is 0.79, while the information flow between the SPI-3 $(t - 2)$ data and the SPI-3 $(t)$ data $(pTE_{SPI3(t-2) \to SPI3(t)})$ is 0.60. Likewise, as an example, the one-phase delay pTE value between the SPI-6 $(t - 1)$ data and the SPI-6 $(t)$ data $(pTE_{SPI6(t-1) \to SPI6(t)}$, is 0.78, while the information flow between the SPI-6 $(t - 2)$ data and the SPI-6 $(t)$ data $(pTE_{SPI6(t-2) \to SPI6(t)})$ is 0.62.

Normalized pTE values between $\varsigma$ time lag SPI-3 $(t - \varsigma)$ data (for $\varsigma = 0,1,2,\ldots,20$) and SPI-3 $(t)$ data and also between $\varsigma$ time lag SPI-6 $(t - \varsigma)$ data (for $\varsigma = 0,1,2,\ldots,20$) and SPI-6 $(t)$ data are shown in Figure 6b.

From Figure 6, when the values of $pTE_{SPI3(t-\varsigma) \to SPI3(t)}$ are examined for $\varsigma > 0$, it is seen that the most information flow is for $pTE_{SPI3(t-1) \to SPI3(t)}$. From here, it can be said that if a 1-input model is to be used in the one ahead estimation of SPI-3 $(t)$ data, the best estimation can be made with SPI-3 $(t - 1)$ input. To give another example, if the values of $pTE_{SPI6(t-\varsigma) \to SPI6(t)}$ for $\varsigma > 0$ are sorted from largest to smallest, then the sort list will be $pTE_{SPI6(t-1) \to SPI6(t)}$, $pTE_{SPI6(t-2) \to SPI6(t)}$, and $pTE_{SPI6(t-5) \to SPI6(t)}$. From this point of view, if we want to use a 3-input model for the one ahead estimation of SPI-6 $(t)$ data, we can use the inputs of SPI-6 $(t - 1)$, SPI-6 $(t - 2)$, and SPI-6 $(t - 5)$ in the study.

The inputs determined for the one ahead forecasting normalized pTE values are given in Table 3.

**Table 3.** Input features defined according to the one-phase delay normalized pTE values for one ahead forecasting.

| Time Scale | Number of Inputs | Inputs | Time Scale | Number of Inputs | Inputs |
|---|---|---|---|---|---|
| 3 months | 1 | $t-1$ | 6 months | 1 | $t-1$ |
| 3 months | 2 | $t-1, t-2$ | 6 months | 2 | $t-1, t-2$ |
| 3 months | 3 | $t-1, t-2, t-3$ | 6 months | 3 | $t-1, t-2, t-5$ |

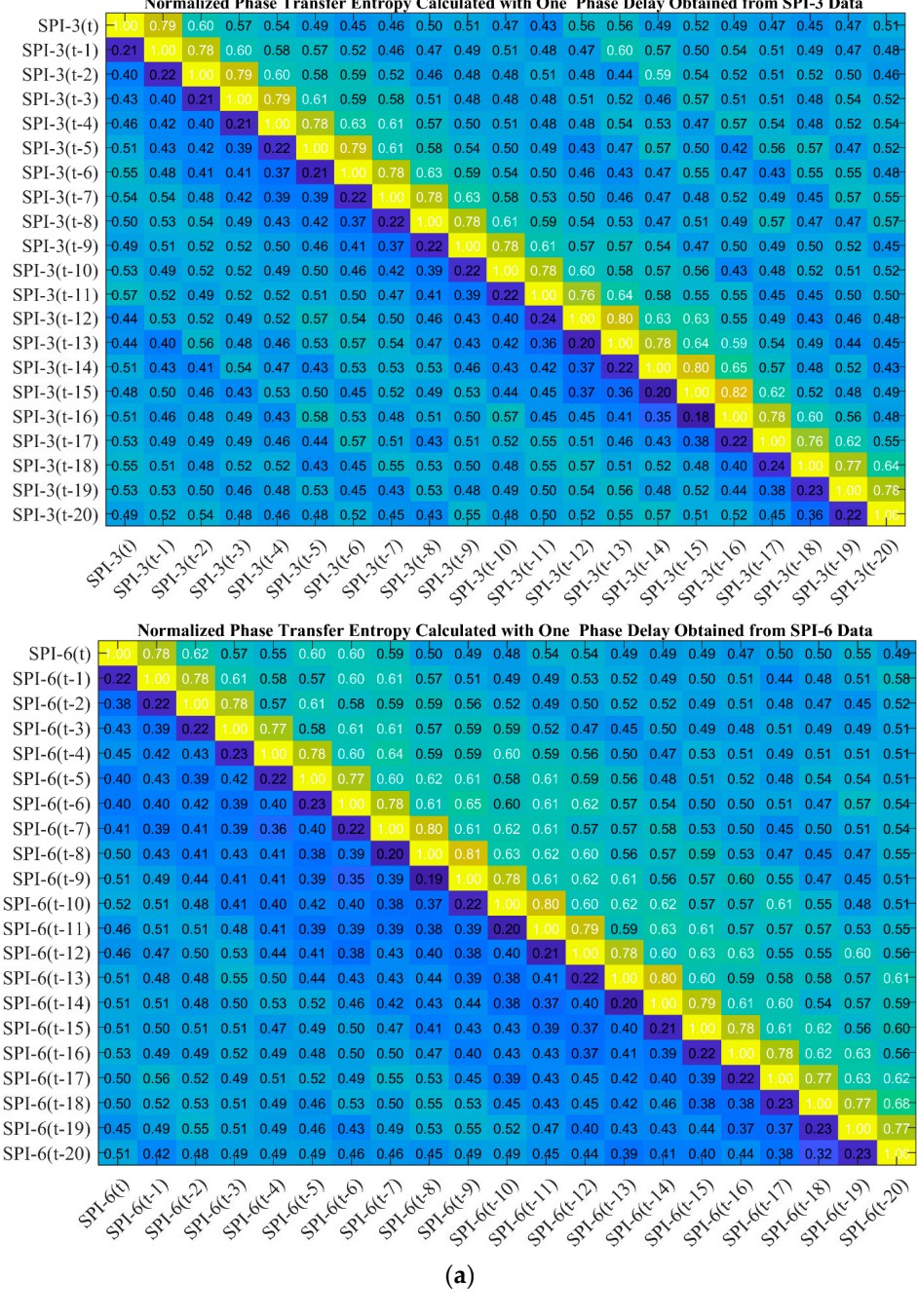

(**a**)

**Figure 6.** *Cont.*

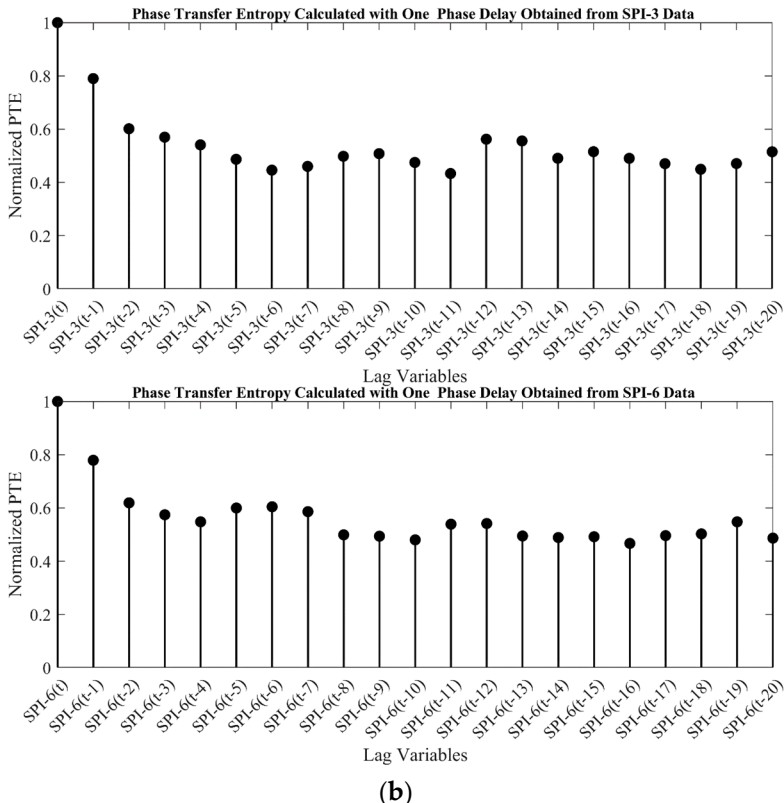

**(b)**

**Figure 6.** Normalized pTE calculated with one-phase delay obtained from SPI-3 and SPI-6 data (**a**) between data with all lag variables and data with all lag variables and (**b**) between data with all lag variables and data with t time indices.

Similar to the approach for determining the inputs and time indices to be used for the one-step ahead prediction model based on the pTE values calculated with a one-phase delay, as seen in Figure 6, the inputs and time indices for the two-step ahead prediction were determined using pTE values calculated with a two-phase delay, and for the three-step ahead prediction, the inputs and time indices were determined using pTE values calculated with a three-phase delay.

The inputs determined for two ahead forecasting using normalized pTE values are given in Table 4.

**Table 4.** Input features defined according to the two-phase delay normalized pTE values for two ahead forecasting.

| Time Scale | Number of Inputs | Inputs | Time Scale | Number of Inputs | Inputs |
|---|---|---|---|---|---|
| 3 months | 1 | $t-2$ | 6 months | 1 | $t-2$ |
| 3 months | 2 | $t-2, t-3$ | 6 months | 2 | $t-2, t-4$ |
| 3 months | 3 | $t-2, t-3, t-4$ | 6 months | 3 | $t-2, t-4, t-5$ |

The inputs determined for three ahead forecasting using normalized pTE values are given in Table 5.

**Table 5.** Input features defined according to the three-phase delay normalized pTE values for three ahead forecasting.

| Time Scale | Number of Inputs | Inputs | Time Scale | Number of Inputs | Inputs |
|---|---|---|---|---|---|
| 3 months | 1 | $t-3$ | 6 months | 1 | $t-3$ |
| 3 months | 2 | $t-3, t-4$ | 6 months | 2 | $t-3, t-4$ |
| 3 months | 3 | $t-3, t-4, t-7$ | 6 months | 3 | $t-3, t-4, t-5$ |

### 2.9.2. GWO-TQWT-ANN, GWO-TQWT-GPR and GWO-TQWT-SVR Combined Forecasting Model Development Phase

In this phase of the study, subband signals were obtained using the TQWT decomposition method. The GWO technique was applied in order to determine the optimum values for the Q and J parameters in the TQWT method. During the optimization of the Q parameter, the parameter value range is selected between 1 and 6. In the same way, the level of decomposition is determined in the range of 1 to 11.

After decomposition of the SPI data, each subband signal was forecasted with ANN in the GWO-TQWT-ANN model. During the construction of the ANN model, one hidden layer was used, and also, the number of neurons in the hidden layer was optimized with the GWO technique. The maximum number of neurons in the hidden layer was determined as 10. Thus, in the GWO-TQWT-ANN model, three parameters: Q, J, and the number of neurons were optimized with the GWO technique.

In the GWO-TQWT-GPR model, which was designed as the second model, the subband signals obtained by TQWT were forecasted using the GPR method. During the development of the GPR model, the Squared Exponential Kernel, Exponential Kernel, Rational Quadratic Kernel, and ARD Squared Exponential Kernel functions were tried as kernel functions. The ARD Squared Exponential Kernel function is used for predicting studies, because it provides better results. In addition, the kernel parameters were optimized using GWO while using the ARD Squared Exponential kernel. Thus, in the GWO-TQWT-GPR model, three parameters: the Q, J, and kernel parameters were optimized with the GWO technique.

In the GWO-TQWT-SVR model, which was designed as the third model, the subband signals obtained by TQWT were forecasted using the SVR method. During the development of the SVR model, the Gaussian Kernel, Radial Basis Kernel, Linear Kernel, and Polynomial Kernel functions were tried as kernel functions. The Radial Basis Kernel function is used for predicting studies, because it provides better results. In addition, the kernel parameters were optimized using GWO while using the Radial Basis kernel. Thus, in the GWO-TQWT-SVR model, three parameters: the Q, J, and kernel parameters were optimized with the GWO technique.

### 2.9.3. EMD-ANN, EMD-GPR, EMD-SVR, VMD-ANN, VMD-GPR, and VMD-SVR Combined Models for Benchmark

At this stage of the study, the performance of the forecasting models developed with preprocessing techniques based on the EMD and VMD techniques was analyzed in order to compare the performance of the model developed with the preprocessing technique based on the TQWT technique.

The SPI training and testing data were decomposed into five subbands (where the stopping criterion was met) using the EMD and VMD methods. Each subband signal obtained using EMD and VMD methods were forecasted using the ANN, GPR, and SVR methods. During the development of the ANN model, one hidden layer was used, the number of neurons in the hidden layer was tested by increasing from 1 to 10, and the number of neurons where the best performance was obtained was used during development of the EMD-ANN and VMD-ANN models.

The ARD Squared Exponential approach was used in the forecasting phase of the EMD and VMD subband signals with the GPR model. The Kernel values were determined by the widely used Fully Independent Conditional (FIC) [61] approach.

The Radial Basis Kernel function was used during the development of the SVR model, and Bayesian Optimization with the support sequential minimal optimization (SMO) approach was used to determine the Kernel parameters.

2.9.4. ANN, GPR, and SVR Stand-Alone Models for Benchmark

In this study, SPI data were forecasted with ANN, SVR, and GPR models without preprocessing. The same approach utilized in the forecasting of EMD and VMD subband signals were used for development of the ANN, GPR, and SVR models in the forecasting study.

## 3. Results

### 3.1. Determination of Input Variables

Phase transfer entropy is a method that helps us gain insights by identifying relationships between data. In this study, for the first time in the literature, a method using the pTE approach was proposed to predict SPI-3 and SPI-6 data based on their previous values. The goal was to determine which past values of SPI data, known as "time lags", should be considered in the prediction process. This determination was made by analyzing the connection and information flow between the data at time $(t - \varsigma)$ and $(t - \tau)$ using the pTE values calculated from the Hankelization matrix of SPI data. As a result, the study aimed to enhance the performance of the prediction model by evaluating the calculated pTE values at different time indices and applying appropriately time-indexed data as inputs to the prediction model.

In this study, one- to three-phase delay transfer entropy values calculated from a matrix obtained by shifting the SPI data in time and the input features to be used for the forecasting models were determined. The connectivity matrix showing the one- to three-phase delay transfer entropy values calculated for forecasting studies is shown in Figure 6 as an example calculated using one-phase delay pTE. As can be seen from the one-phase delay PTE data using the SPI-3 data, the first three largest information flows are between SPI-3 (t) and SPI-3 (t − 1), SPI-3 (t − 2), and SPI-3 (t − 3) data. Therefore, SPI (t − 1), SPI (t − 2), and SPI (t − 3) data are used in the one ahead forecasting of SPI-3 (t) data. Additionally, as the first three big two-phase delayed pTE values are obtained in the t − 2, t − 3, and t − 4 time indices, it is proposed to use SPI-3 (t − 2), SPI-3 (t − 3), and SPI-3 (t − 4) data in the forecasting of SPI-3 (t) data. Also, as the first three big three-phase delayed pTE values are obtained in the t − 3, t − 4, and t − 7 time indices, it is proposed to use SPI-3 (t − 3), SPI-3 (t − 4), and SPI-3 (t − 7) data in forecasting SPI-3 (t) data. Similarly, it is suggested to use SPI-6 (t − 1), SPI-6 (t − 2), and SPI-6 (t − 5) data as input variables for the one ahead forecasting of SPI-6 (t) data according to the one-phase delay pTE values. It is suggested to use SPI-6 (t − 2), SPI-6 (t − 4), and SPI-6 (t − 5) data as the input variables for two ahead forecasting according to the two-phase delay pTE values. With the same approach, it is suggested to use SPI-6 (t − 3), SPI-6 (t − 4), and SPI-6 (t − 5) data for the three ahead predictions of SPI-6 (t) data according to the three three-phase delayed pTE values. In order to analyze the effects of the input variables on the forecasting performance, instead of the inputs we determined with pTE, SPI-3, and SPI-6 data, were estimated them with sequentially different inputs. Table 6 displays the prediction results obtained using inputs determined by pTE and inputs applied with consecutive time lags.

For the one-ahead forecasting of SPI-6 data, for example, SPI-6 (t − 1), SPI-6 (t − 2), and SPI-6 (t − 3) inputs are used as inputs for the ANN, GPR, and SVR models. However, a higher performance was obtained in the forecasting study using the SPI-6 (t − 1), SPI-6 (t − 2), and SPI-6 (t − 5) inputs recommended in this study for the forecasting of SPI-6 (t) data. According to these results, it can be observed that, when inputs corresponding to the time indices determined by the proposed pTE-based approach are applied, the prediction performance is better than when inputs corresponding to consecutive time indices are applied.

**Table 6.** The prediction performances of models where inputs corresponding to the recommended method's determined time indices are applied and models where inputs corresponding to consecutive time indices are applied.

| Three Inputs—One Ahead Forecasting | | | | | | | |
|---|---|---|---|---|---|---|---|
| **SPI-6 (Three Inputs (t − 1, t − 2, t − 3))** | | | | **SPI-6 (Three Inputs (t − 1, t − 2, t − 5), Defined by Proposed Method)** | | | |
| **Models** | **MSE** | **MAE** | **R** | **R$^2$** | **MSE** | **MAE** | **R** | **R$^2$** |
| ANN | 0.2957 | 0.4153 | 0.8157 | 0.6653 | 0.2800 | 0.4048 | 0.8271 | 0.6842 |
| GPR | 1.2777 | 0.6065 | 0.8129 | 0.6607 | 0.2832 | 0.4012 | 0.8241 | 0.6792 |
| SVR | 1.5763 | 1.0088 | 0.8177 | 0.6684 | 1.5623 | 1.0056 | 0.8245 | 0.6797 |
| **Three Inputs—Two Ahead Forecasting** | | | | | | | |
| **SPI-6 (Three Inputs (t − 2, t − 3, t − 4))** | | | | **SPI-6 (Three Inputs (t − 2, t − 4, t − 5), Defined by Proposed Method)** | | | |
| **Models** | **MSE** | **MAE** | **R** | **R$^2$** | **MSE** | **MAE** | **R** | **R$^2$** |
| ANN | 0.4983 | 0.5637 | 0.6677 | 0.4458 | 0.4844 | 0.5561 | 0.6761 | 0.4571 |
| GPR | 1.3651 | 0.9464 | 0.6689 | 0.4474 | 1.3508 | 0.9410 | 0.6692 | 0.4479 |
| SVR | 1.4105 | 0.9584 | 0.6660 | 0.4435 | 1.3793 | 0.9478 | 0.6713 | 0.4507 |
| **Three Inputs—Three Ahead Forecasting** | | | | | | | |
| **SPI-3 (Three Inputs (t − 3, t − 4, t − 5))** | | | | **SPI-3 (Three Inputs (t − 3, t − 4, t − 7), Defined by Proposed Method)** | | | |
| **Models** | **MSE** | **MAE** | **R** | **R$^2$** | **MSE** | **MAE** | **R** | **R$^2$** |
| ANN | 0.8767 | 0.7351 | −0.025 | 0.0007 | 0.8520 | 0.7106 | 0.1013 | 0.0103 |
| GPR | 0.8855 | 0.7422 | 0.1041 | 0.0108 | 0.8490 | 0.7205 | 0.1345 | 0.0181 |
| SVR | 1.1638 | 0.8726 | 0.4998 | 0.2498 | 1.1271 | 0.8570 | 0.5060 | 0.2560 |

*3.2. Computational Results for the SPI Forecasting Study*

In this study, SPI-3 and SPI-6 data were estimated with a new approach based on GWO, TQWT, and a machine learning model. ANN, GPR, and SVR models from the machine learning techniques were used. Thus, the drought forecasting performances of the new approach with the GWO-TQWT-ANN, GWO-TQWT-GPR, and GWO-TQWT-SVR models were analyzed. In order to compare the performance of this TQWT preprocessing approach, EMD and VMD preprocessing methods and forecasting models, which are widely used in the literature, were developed. One to three input one ahead forecasting study of SPI-3 and SPI-6 data were performed using the EMD-ANN, EMD-GPR, EMD-SVR, VMD-ANN, VMD-GPR, and VMD-SVR methods. In addition, 1–3 input one ahead forecasting of SPI-3 and SPI-6 data were performed with the ANN, GPR, and SVR models. The MSE, MAE, R, and R$^2$ parameters were calculated to analyze the forecasting performances of all the models. The following table shows the performances obtained from all the models.

As seen in Table 7, the performances of the proposed GWO-TQWT-ML models in the one ahead forecasting of SPI-3 and SPI-6 data are outstanding. It is seen that the performance of the GWO-TQWT-GPR model is slightly better than the GWO-TQWT-ANN and GWO-TQWT-SVR models. To see the model performances graphically, scatter plots for the one ahead forecasting models are shown in Figure 7.

**Table 7.** Performance parameters obtained from the one, two, and three input one ahead forecasting models.

| | **One Input—One Ahead Forecasting** | | | | | | | |
|---|---|---|---|---|---|---|---|---|
| | **SPI-3** | | | | **SPI-6** | | | |
| **Models** | **MSE** | **MAE** | **R** | **R²** | **MSE** | **MAE** | **R** | **R²** |
| ANN | 0.4349 | 0.5254 | 0.7061 | 0.4986 | 0.2933 | 0.4220 | 0.8200 | 0.6724 |
| GPR | 1.2899 | 0.9112 | 0.6989 | 0.4885 | 1.5511 | 1.0035 | 0.8130 | 0.6609 |
| SVR | 1.3892 | 0.9420 | 0.6998 | 0.4898 | 1.5921 | 1.0163 | 0.8133 | 0.6615 |
| EMD-ANN | 0.3650 | 0.4663 | 0.7614 | 0.5797 | 0.2180 | 0.3652 | 0.8636 | 0.7458 |
| EMD-GPR | 0.3620 | 0.4602 | 0.7647 | 0.5847 | 0.2228 | 0.3676 | 0.8604 | 0.7403 |
| EMD-SVR | 0.3953 | 0.4852 | 0.7398 | 0.5473 | 0.2523 | 0.3983 | 0.8403 | 0.7061 |
| VMD-ANN | 0.6114 | 0.6370 | 0.6242 | 0.3897 | 0.3017 | 0.4254 | 0.8193 | 0.6712 |
| VMD-GPR | 0.6261 | 0.6445 | 0.6168 | 0.3805 | 0.3092 | 0.4292 | 0.8155 | 0.6650 |
| VMD-SVR | 0.6361 | 0.6473 | 0.6109 | 0.3732 | 0.3068 | 0.4286 | 0.8174 | 0.6681 |
| GWO-TQWT-ANN | 0.3327 | 0.4486 | 0.7900 | 0.6242 | 0.2150 | 0.3600 | 0.8723 | 0.7610 |
| GWO-TQWT-GPR | 0.2768 | 0.4110 | 0.8249 | 0.6805 | 0.1747 | 0.3343 | 0.8966 | 0.8039 |
| GWO-TQWT-SVR | 0.2938 | 0.4269 | 0.8136 | 0.6620 | 0.1769 | 0.3422 | 0.8956 | 0.8020 |
| | **Two Inputs—One Ahead Forecasting** | | | | | | | |
| | **SPI-3** | | | | **SPI-6** | | | |
| **Models** | **MSE** | **MAE** | **R** | **R²** | **MSE** | **MAE** | **R** | **R²** |
| ANN | 0.3962 | 0.5167 | 0.7389 | 0.5460 | 0.2897 | 0.4103 | 0.8223 | 0.6761 |
| GPR | 1.3398 | 0.9260 | 0.7270 | 0.5285 | 1.5708 | 1.0099 | 0.8215 | 0.6748 |
| SVR | 1.3750 | 0.9370 | 0.7236 | 0.5235 | 1.5669 | 1.0074 | 0.8218 | 0.6753 |
| EMD-ANN | 0.2360 | 0.3661 | 0.8556 | 0.7320 | 0.1062 | 0.2444 | 0.9364 | 0.8768 |
| EMD-GPR | 0.2682 | 0.4023 | 0.8342 | 0.6958 | 0.1084 | 0.2434 | 0.9345 | 0.8733 |
| EMD-SVR | 0.2946 | 0.4319 | 0.8199 | 0.6722 | 0.1612 | 0.3021 | 0.9012 | 0.8122 |
| VMD-ANN | 1.0359 | 0.8302 | 0.5611 | 0.3148 | 0.3108 | 0.4446 | 0.8564 | 0.7334 |
| VMD-GPR | 1.4065 | 0.9264 | 0.4948 | 0.2448 | 0.4260 | 0.5103 | 0.8233 | 0.6778 |
| VMD-SVR | 1.3963 | 0.9231 | 0.4955 | 0.2455 | 0.4297 | 0.5112 | 0.8188 | 0.6705 |
| GWO-TQWT-ANN | 0.0032 | 0.0447 | 0.9982 | 0.9964 | 0.0023 | 0.0372 | 0.9987 | 0.9975 |
| GWO-TQWT-GPR | 0.0017 | 0.0328 | 0.9991 | 0.9982 | 0.0012 | 0.0265 | 0.9993 | 0.9987 |
| GWO-TQWT-SVR | 0.0167 | 0.1019 | 0.9909 | 0.9820 | 0.0083 | 0.0796 | 0.9960 | 0.9920 |
| | **Three Inputs—One Ahead Forecasting** | | | | | | | |
| | **SPI-3** | | | | **SPI-6** | | | |
| **Models** | **MSE** | **MAE** | **R** | **R²** | **MSE** | **MAE** | **R** | **R²** |
| ANN | 0.3749 | 0.4901 | 0.7529 | 0.5669 | 0.2800 | 0.4048 | 0.8271 | 0.6842 |
| GPR | 1.3546 | 0.9311 | 0.7527 | 0.5666 | 0.2832 | 0.4012 | 0.8241 | 0.6792 |
| SVR | 1.3460 | 0.9267 | 0.7474 | 0.5586 | 1.5623 | 1.0056 | 0.8245 | 0.6797 |
| EMD-ANN | 0.2209 | 0.3601 | 0.8672 | 0.7520 | 0.1036 | 0.2542 | 0.9365 | 0.8771 |
| EMD-GPR | 0.2453 | 0.3726 | 0.8498 | 0.7222 | 0.1065 | 0.2574 | 0.9336 | 0.8716 |
| EMD-SVR | 0.2645 | 0.3923 | 0.8335 | 0.6948 | 0.1581 | 0.3066 | 0.8997 | 0.8094 |
| VMD-ANN | 1.2388 | 0.8845 | 0.5168 | 0.2671 | 0.3077 | 0.4044 | 0.8513 | 0.7247 |
| VMD-GPR | 2.6403 | 1.2710 | 0.3396 | 0.1153 | 0.6311 | 0.6252 | 0.7708 | 0.5942 |
| VMD-SVR | 1.7225 | 1.0289 | 0.4554 | 0.2074 | 0.6040 | 0.6068 | 0.7660 | 0.5868 |
| GWO-TQWT-ANN | 0.0024 | 0.0351 | 0.9987 | 0.9975 | 0.0014 | 0.0276 | 0.9992 | 0.9985 |
| GWO-TQWT-GPR | 0.0004 | 0.0153 | 0.9998 | 0.9996 | 0.0002 | 0.0115 | 0.9999 | 0.9998 |
| GWO-TQWT-SVR | 0.0043 | 0.0554 | 0.9977 | 0.9954 | 0.0058 | 0.0641 | 0.9970 | 0.9940 |

In this study, two and three ahead forecasting of SPI data were also performed with the GWO-TQWT-ML approach. The two and three ahead forecasting performances of the GWO-TQWT-ANN, GWO-TQWT-GPR, and GWO-TQWT-SVR models are shown in Table 8. Also, scatter diagrams are shown in Figures 8–11.

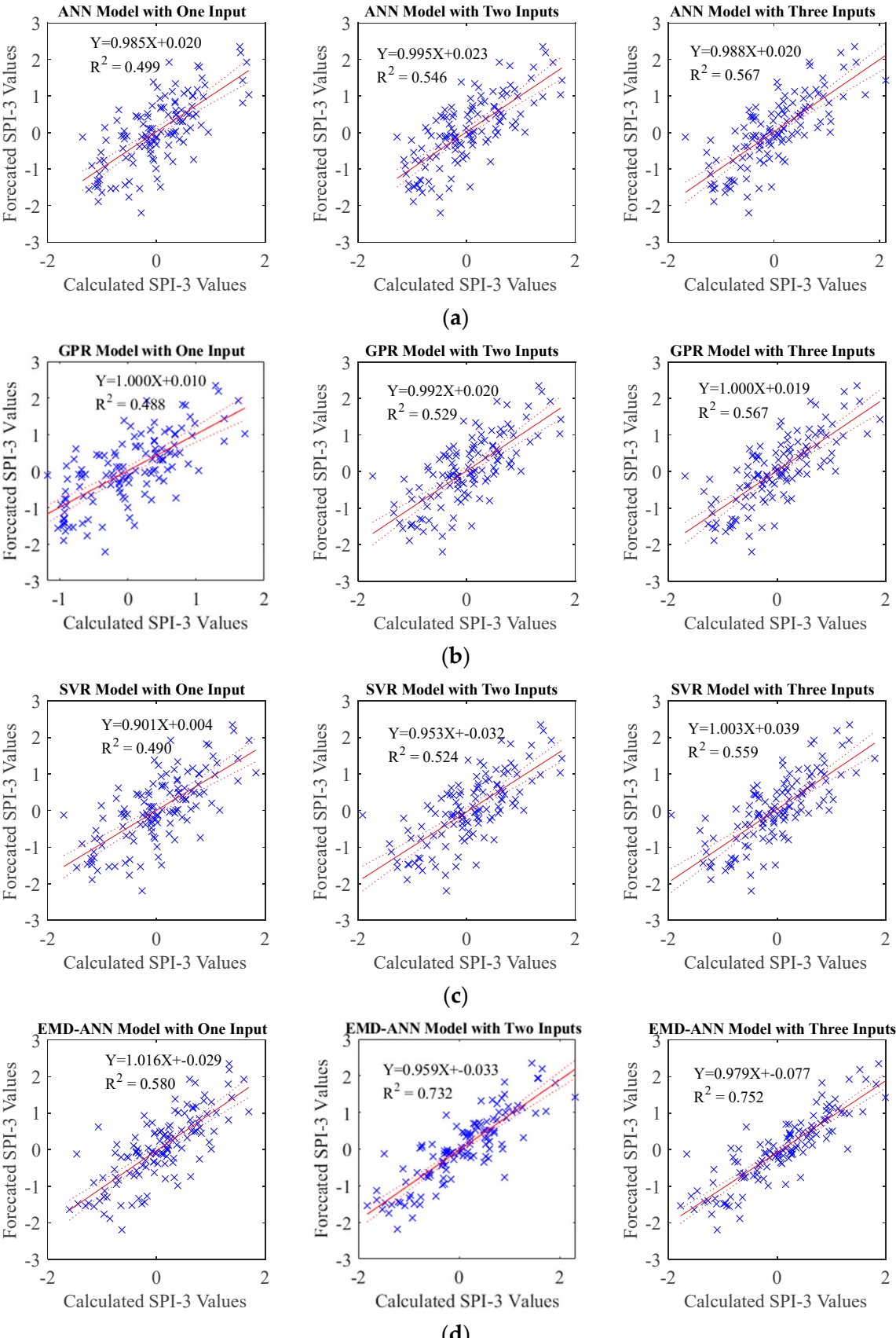

**Figure 7.** *Cont.*

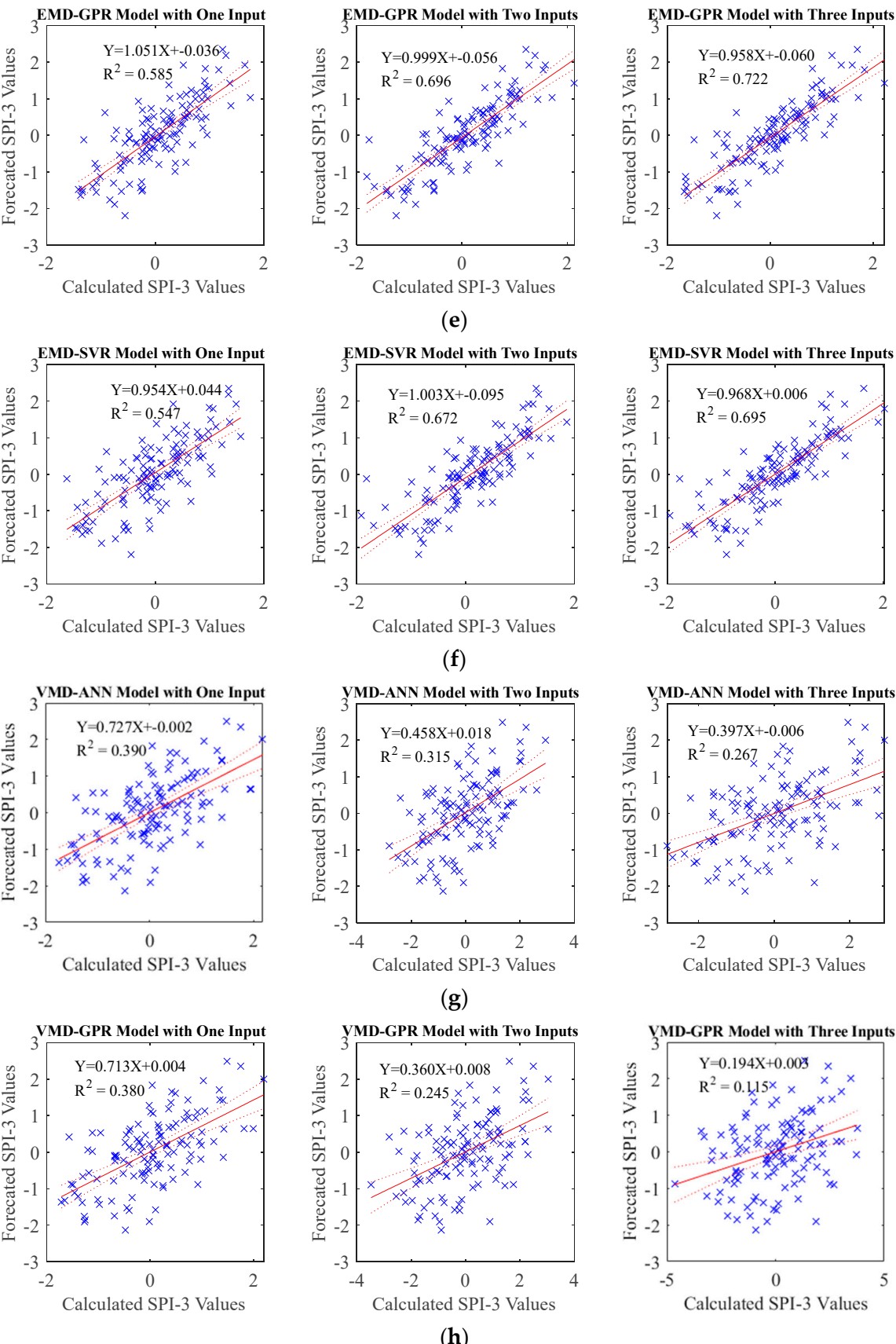

**Figure 7.** *Cont.*

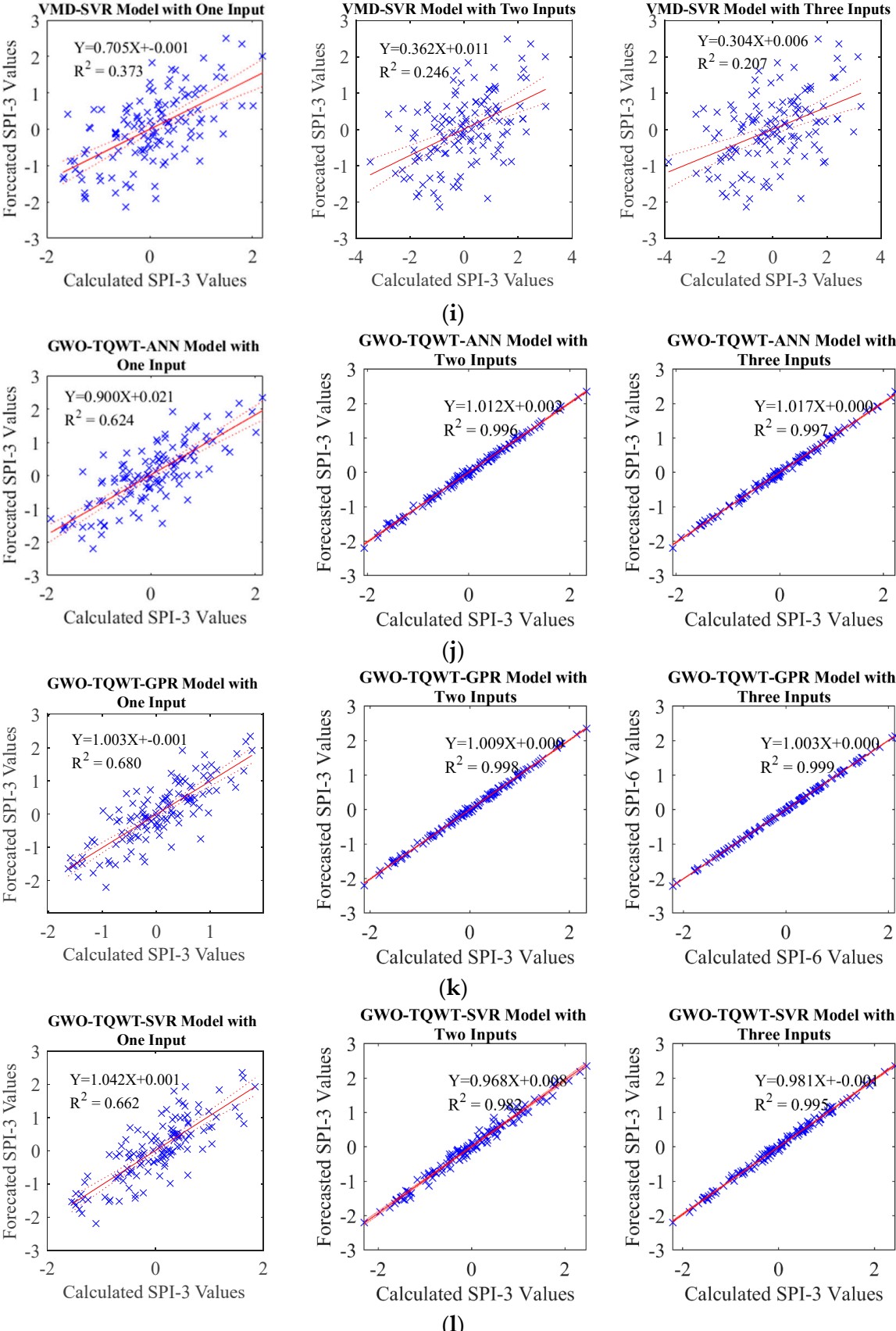

**Figure 7.** *Cont.*

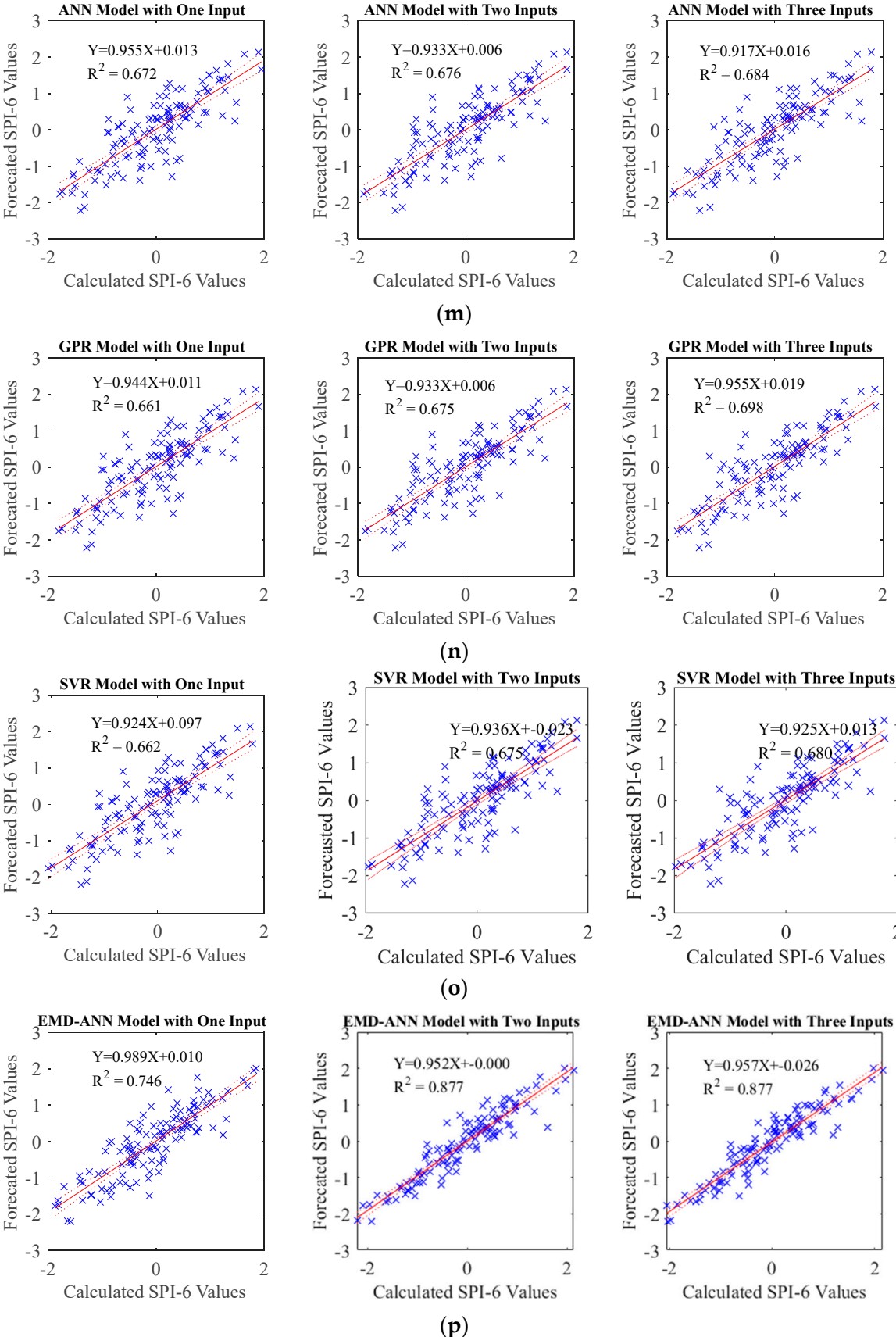

**Figure 7.** *Cont.*

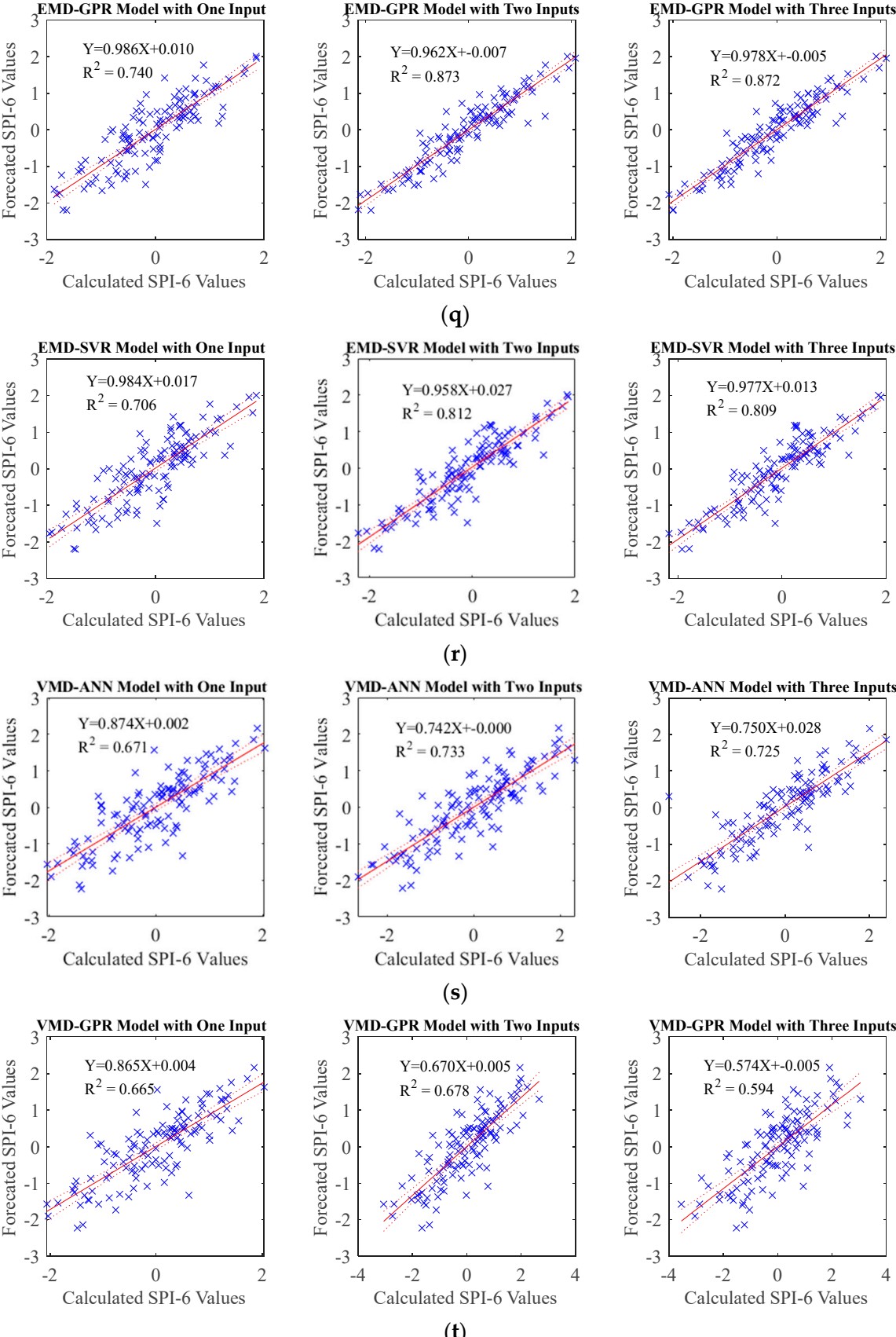

**Figure 7.** *Cont.*

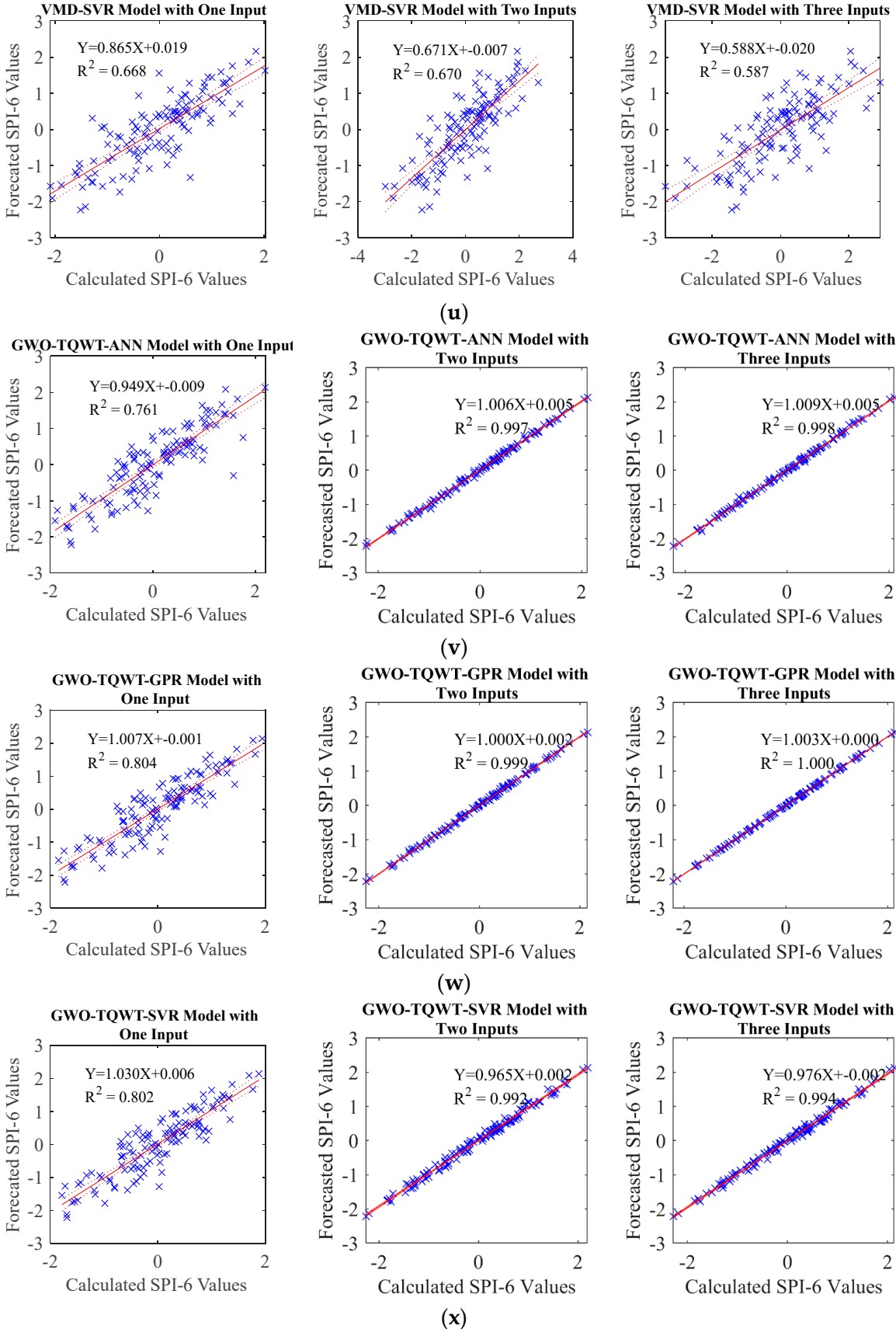

**Figure 7.** Scatter plots obtained from the (**a**–**l**) one ahead forecasted SPI-3 data and (**m**–**x**) one ahead forecasted SPI-6 data.

**Table 8.** Performance parameters obtained from the one to three input two and three ahead forecasting models.

| Models | SPI-3 | | | | SPI-6 | | | |
|---|---|---|---|---|---|---|---|---|
| | MSE | MAE | R | $R^2$ | MSE | MAE | R | $R^2$ |
| Two Inputs Model | | | | | | | | |
| GWO-TQWT-ANN | 0.0917 | 0.1527 | 0.9478 | 0.8983 | 0.0128 | 0.0841 | 0.9926 | 0.9853 |
| GWO-TQWT-GPR | 0.0138 | 0.0942 | 0.9921 | 0.9843 | 0.0082 | 0.0699 | 0.9953 | 0.9907 |
| GWO-TQWT-SVR | 0.0975 | 0.2585 | 0.9445 | 0.8920 | 0.0918 | 0.2512 | 0.9518 | 0.9060 |
| Three Inputs Model | | | | | | | | |
| GWO-TQWT-ANN | 0.0036 | 0.0475 | 0.9981 | 0.9962 | 0.0095 | 0.0742 | 0.9949 | 0.9897 |
| GWO-TQWT-GPR | 0.0034 | 0.0464 | 0.9982 | 0.9963 | 0.0023 | 0.0379 | 0.9988 | 0.9975 |
| GWO-TQWT-SVR | 0.0307 | 0.1415 | 0.9821 | 0.9646 | 0.0331 | 0.1482 | 0.9810 | 0.9624 |
| | SPI-3 | | | | SPI-6 | | | |
| Models | MSE | MAE | R | $R^2$ | MSE | MAE | R | $R^2$ |
| Two Inputs Model | | | | | | | | |
| GWO-TQWT-ANN | 0.0846 | 0.2085 | 0.9495 | 0.9016 | 0.0272 | 0.1320 | 0.9846 | 0.9694 |
| GWO-TQWT-GPR | 0.0399 | 0.1582 | 0.9765 | 0.9535 | 0.0248 | 0.1230 | 0.9860 | 0.9722 |
| GWO-TQWT-SVR | 0.2499 | 0.4179 | 0.8478 | 0.7187 | 0.1822 | 0.3540 | 0.8970 | 0.8045 |
| Three Inputs Model | | | | | | | | |
| GWO-TQWT-ANN | 0.0096 | 0.0794 | 0.9945 | 0.9891 | 0.0256 | 0.1323 | 0.9856 | 0.9713 |
| GWO-TQWT-GPR | 0.0152 | 0.1002 | 0.9913 | 0.9826 | 0.0108 | 0.0854 | 0.9941 | 0.9882 |
| GWO-TQWT-SVR | 0.1563 | 0.3169 | 0.9110 | 0.8298 | 0.1390 | 0.3038 | 0.9183 | 0.8432 |

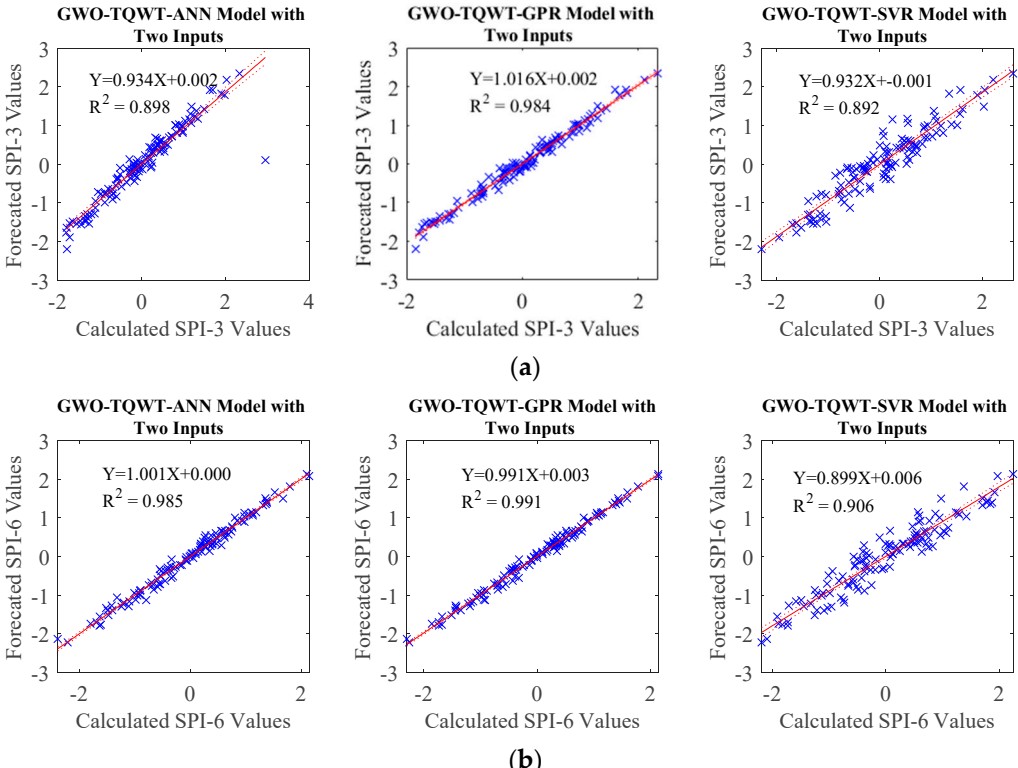

**Figure 8.** Scatter plots obtained from the (**a**) two ahead forecasted SPI-3 data using two inputs and (**b**) two ahead forecasted SPI-6 data using two inputs.

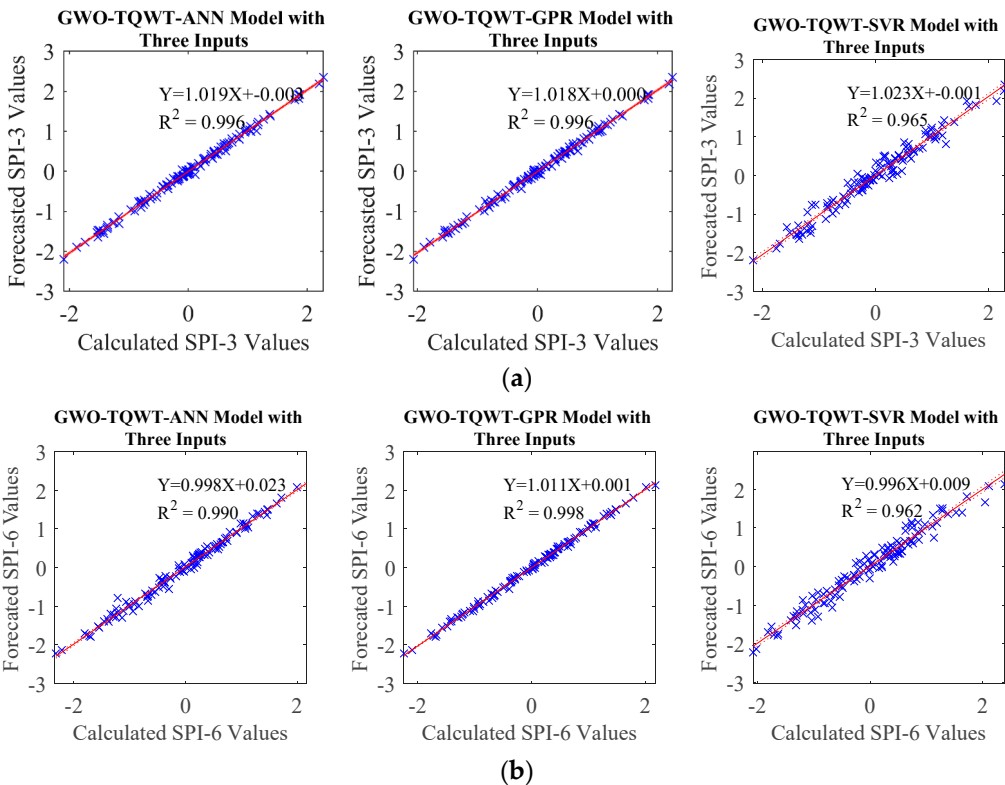

**Figure 9.** Scatter plots obtained from the (**a**) two ahead forecasted SPI-3 data using three inputs and (**b**) two ahead forecasted SPI-6 data using three inputs.

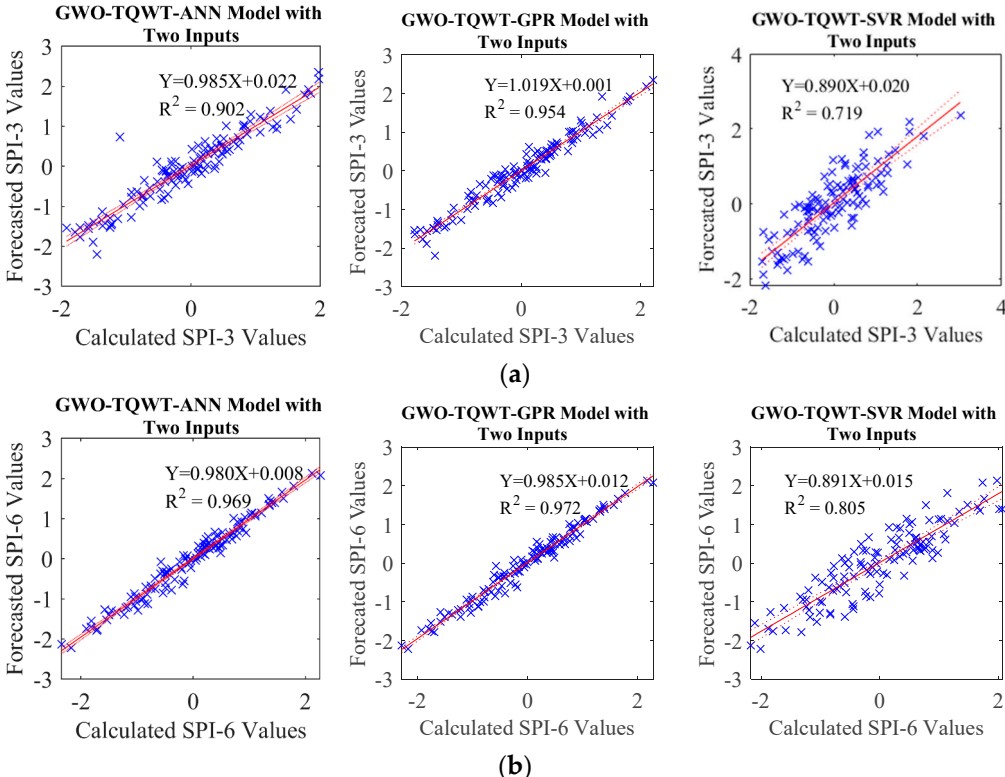

**Figure 10.** Scatter plots obtained from the (**a**) three ahead forecasted SPI-3 data using two inputs and (**b**) three ahead forecasted SPI-6 data using two inputs.

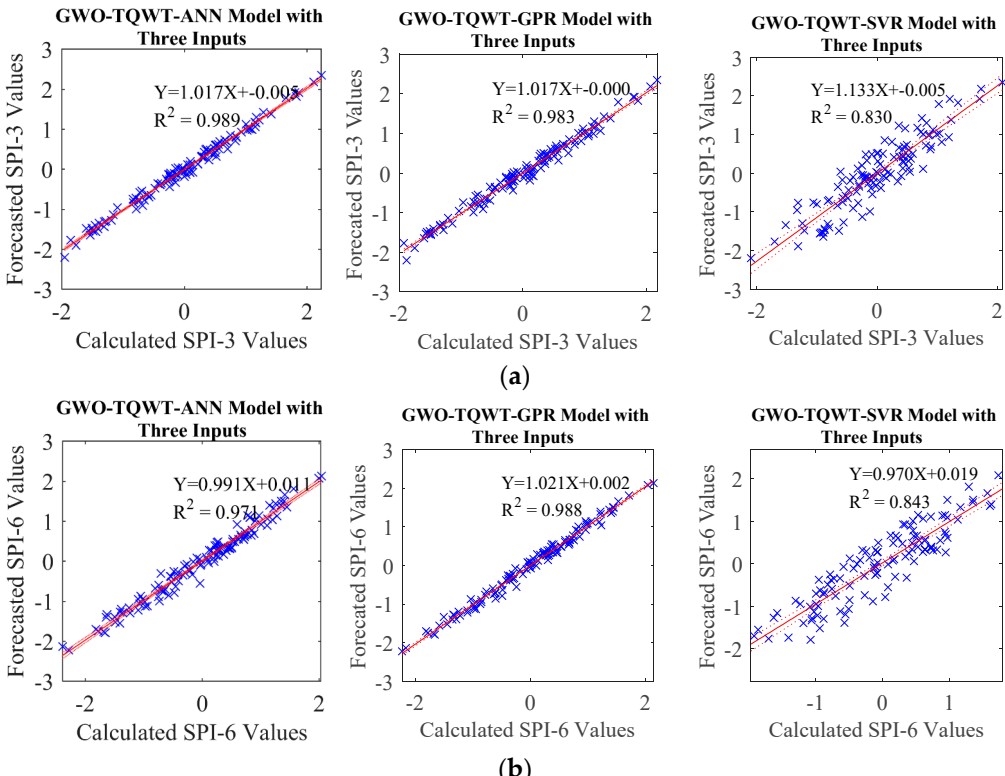

**Figure 11.** Scatter plots obtained from the (**a**) three ahead forecasted SPI-3 data using three inputs and (**b**) three ahead forecasted SPI-6 data using three inputs.

## 4. Discussion and Conclusions

The importance of SPI prediction lies in its ability to provide valuable insights into future precipitation patterns, aiding in water resource management, drought monitoring, and climate adaptation efforts. In the literature, there have been many methods that use the machine learning (ML) approach for high-performance predictions of the SPI in recent years. A comprehensive review study on these methods can be found in the references. As observed in the literature [62], ML methods have proven to be highly effective in SPI prediction, yielding high-performance results. In this study, as a new approach compared to previous works, the optimal time lag value for SPI prediction using the past values of SPI data has been determined using phase transfer entropy (pTE). Additionally, a novel pTE-Grey Wolf Optimization (GWO)-Tunable Q Wavelet Transform (TQWT)-ML method, utilizing preprocessed data with parameters optimized by GWO and the TQWT technique, has been proposed to enhance the performance of ML methods in SPI prediction.

In this study, a high-performance SPI drought index estimation model was developed with an end-to-end approach. In this method, a connectivity analysis was performed for the first time in the literature for SPI data. The phase transfer entropy method was used for the connectivity analysis, and the interconnected time indices of the SPI data were obtained. Thus, the most important lag time for the SPI input data for the forecasting models was determined.

As can be seen in Figure 6 and Tables 3–5, the input data with the highest information flow for a certain time index that can be applied to forecasting models were obtained for SPI-3 and SPI-6 data. As seen in Table 6, when the method utilizing the recommended pTE approach with a specified time lag for SPI data is compared to the prediction results using consecutive time lag SPI data, it is evident that the prediction performance using input indices from the recommended method is superior.

In the TQWT and ML models, those parameters were optimized with GWO and employed in developing forecasting models. During this process, the SPI data underwent

decomposition into subband signals using TQWT, and subsequently, these derived subband signals were predicted using ML algorithms, specifically Artificial Neural Network (ANN), support vector machine (SVM), and Gaussian Process Regression (GPR). In order to compare the performance of the GWO-TQWT-ML model, ML-based approaches undergoing Empirical Mode Decomposition and Variational Mode Decomposition preprocessing and stand-alone models (ANN, SVR, and GPR) in which the SPI data itself were used were estimated. As a result of these studies, it is seen that the performance of the GWO-TQWT-ML models is quite good in the one ahead forecasting of SPI data, as shown in Table 7, and $R^2$ values up to 0.999 are obtained. From the scatter plots (Figure 7), it is seen that the performance of the proposed model is significantly superior to the other models. In addition, two and three ahead estimation GWO-TQWT-ML models of SPI data were performed. Among the GWO-TQWT-ML models, it can be seen from Table 8 and Figures 8–11 that the performance of the GWO-TQWT-GPR model was slightly better than the prediction performance of the GWO-TQWT-ANN and GWO-TQWT-SVR models.

The key results that address these contributions are as follows:

The introduction of a novel method incorporating time-shifting processes and the phase transfer entropy approach for analyzing connectivity in time series data, marking the first use of this approach in the field, thus the identification of input data for SPI estimations using connectivity values derived from this method.

The utilization of TQWT subband signals in SPI estimation, which enhances the accuracy of predictions.

The optimization of TQWT and ML model parameters through the GWO, improving the overall performance of the forecasting models.

Development of the pTE-GWO-TQWT-GPR model, which demonstrated the highest forecasting performance for SPI data used in drought forecasting.

The potential applicability of the proposed approach not only for SPI data but also for estimating other time series related to hydrological processes and forecasting studies.

These results collectively contribute to the advancement of methodologies for time series analysis and forecasting, particularly in the context of drought prediction and hydrological processes.

**Author Contributions:** Conceptualization, L.L. and M.Ö.; methodology, L.L.; software, L.L.; resources, M.Ö.; and writing—review and editing, L.L. and M.Ö. All authors have read and agreed to the published version of the manuscript.

**Funding:** This research received no external funding.

**Institutional Review Board Statement:** Not applicable.

**Informed Consent Statement:** Not applicable.

**Data Availability Statement:** The data supporting the findings of this study are available upon request from the authors following the publication of the article. The authors may be contacted for data sharing inquiries, and the data can be made available as needed. Additionally, the dataset for this study can be accessed at the following address: https://ral.ucar.edu/solutions/products/camels, accessed on 1 February 2023.

**Conflicts of Interest:** The authors declare no conflict of interest.

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
