# Peer review of "A Novel Approach for High-Performance Estimation of SPI Data in Drought Prediction"

_sustainability, doi:10.3390/su151914046_

Round 1

Reviewer 1 Report

1. The abstract mentions the superior performance of the proposed models, but specific quantitative results should be provided to support this claim.

2. The paper provides a comprehensive overview of drought types, causes, and impacts. However, the writing can be further improved for clarity and organization. Consider breaking down complex sentences into smaller ones. The first paragraph is too long, it is recommended to split it into two paragraphs.

3. The literature review is thorough, but it could benefit from more recent references to reflect the latest advancements in the field. Make sure to include references from the past few years to provide a well-rounded perspective.

4. The usage of station ID 13340000 for calculating SPI-3 and SPI-6 indices is well-described. However, please provide additional context regarding the geographical relevance of this station to the study area. Explain how the station's location relates to the Orofino region and its representativeness for the broader study objectives.

Author Response

Dear Reviewer,

We appreciate your valuable feedback and constructive comments on our manuscript. Your insights have greatly contributed to improving the quality and clarity of our work. In response to your comments, we have made several revisions to the manuscript, which are detailed below.

  1. The abstract mentions the superior performance of the proposed models, but specific quantitative results should be provided to support this claim.

Response: As per your recommendation, the R2 values for the prediction performance obtained with the proposed novel pTE-GWO-TQWT-GPR model for SPI forecasting have been added to the following paragraph, as shown below.

The pTE-GWO-TQWT-GPR model yielded determination coefficient (R2) values for SPI-6 data as follows: 0.8039 for one-input, 0.9987 for two-input, 0.9998 for three-input one ahead prediction respectively, 0.9907 for two-input two ahead prediction, and 0.9722 for two-input three ahead prediction. For SPI-3 data, using the pTE-GWO-TQWT-GPR model, the R2 values were as follows: 0.6805 for one-input, 0.9982 for two-input, 0.9996 for three-input one ahead prediction, 0.9843 for two-input two ahead prediction, 0.9535 for two-input three ahead prediction, 0.9963 for three-input two ahead prediction, and 0.9826 for three-input three ahead prediction.

  1. The paper provides a comprehensive overview of drought types, causes, and impacts. However, the writing can be further improved for clarity and organization. Consider breaking down complex sentences into smaller ones. The first paragraph is too long, it is recommended to split it into two paragraphs.

Response: On the referee's recommendations  the paper has been reorganized for better clarity and understanding. Sentences have been made more straightforward, and the first paragraph has been split into two separate paragraphs as recommended.

  1. The literature review is thorough, but it could benefit from more recent references to reflect the latest advancements in the field. Make sure to include references from the past few years to provide a well-rounded perspective.

Response: New references have been added to the paper, and updates have been made in accordance with your recommendation.

  1. The usage of station ID 13340000 for calculating SPI-3 and SPI-6 indices is well-described. However, please provide additional context regarding the geographical relevance of this station to the study area. Explain how the station's location relates to the Orofino region and its representativeness for the broader study objectives.

Response: In this study, our primary objective is to develop a novel and high-performance model for SPI data prediction. We utilized data from the Orofino region as the application dataset, considering its relevance to our immediate research goals. Orofino's geographical proximity to our study area and the availability of long-term data records from the Orofino station were key factors in our choice. However, in future research, this validated model can be employed for prediction tasks using data from different regions, enabling regional forecasting studies.

The "Study Area and Data" section of the paper has been revised, and the following sections have been added:

In this study, precipitation data was utilized from the Catchment Attributes and Meteorology for Large-sample Studies (CAMELS) dataset [32]–[35]. The CAMELS dataset comprises observed runoff levels and catchment-aggregated meteorological forcing data at the daily timescale. Daily meteorological data were computed using gridded data sources, including the Daymet data [36], the Maurer data [37], and the National Land Data Assimilation System (NLDAS) data [38] in CAMELS dataset. Interested readers can find comprehensive information about this dataset [32]–[35].

For the proposed drought forecasting study, the Daymet data due to its high spatial resolution, is crucial for accurately estimating spatial variability in basins with complex topography. You can access these parameters freely at https://ral.ucar.edu/solutions/products/camels [32], [33]In this study, the SPI-3 and SPI-6 indices were computed using precipitation data collected from the gauge with ID 13340000 from the CAMELS dataset , spanning from October 1, 1980, to December 31, 2014. This gauge is situated in the Clearwater River basin in Orofino, characterized by a substantial drainage area of 14268.92 km2 and geographic coordinates at Latitude  46°28'42", Longitude 116°15'27". The study area is seen in Figure 2.

The precipitation data in this database are daily data and monthly averages of these data are taken as it is seen in Figure 3.

Figure 2. Study Area [39].

We believe that these revisions have significantly strengthened the manuscript, and we are confident that the revised version of the paper is now well-prepared for publication. We would like to express our gratitude for your time and effort in reviewing our work.

Thank you once again for your valuable feedback and guidance.

Sincerely,

Assoc. Prof. Dr. Levent LatifoÄŸlu

Erciyes University

Kayseri/Türkiye

Reviewer 2 Report

Dear authors,

Thank you for this study. The integration of Phase Transfer Entropy (pTE) to identify the most relevant time lags presents an innovative approach, which I appreciate.

However, I believe the strength of your study lies in this novel application of pTE. To truly showcase its utility, I recommend focusing more on contrasting your results from the chosen lags, against sequentially chosen lags. This would not only validate the use of pTE but also clarify its advantage over more conventional techniques. While I understand the desire to present a comprehensive study, attempting to address multiple objectives, some of which have been well-covered in existing literature,  made this study hard to follow. 

Furthermore, there are sections in the manuscript that contain redundant information, making it lengthy and, at times, make it challenging to ascertain the core objective of the study. I have attached a document with specific comments and suggestions for revision.

I look forward to receiving a revised version that focuses on the novel aspect of this work and delineates it clearly from conventional methodologies.

Best wishes,

Reviewer 

Author Response

Dear Reviewer,

We appreciate your valuable feedback and constructive comments on our manuscript. Your insights have greatly contributed to improving the quality and clarity of our work. In response to your comments, we have made several revisions to the manuscript, which are detailed below.

  •  

 Response:

As per your recommendation, I removed the abbreviations that appear only once in the abstract.

  •  

Response:  GWO was removed from the keywords.

  •  

Response: The word it has been replaced with drought

  •  

Response: The mentioned article is referenced within the article

  •  

Response: Thank you for your suggestion. The specified paragraph and article have been rewritten with abbreviations in many places.

  •  

Response: It was written as a new paragraph

  •  

Response: That sentence was rewritten as seen in below.

“In a previous study by Mokhtarzad et al., drought prediction was conducted using the Standardized Precipitation Index (SPI) with three different modeling techniques: artificial neural network (ANN), support vector machine (SVM), and adaptive neu-ro-fuzzy interface system (ANFIS). Notably, the SVM model outperformed both the ANN and ANFIS models in terms of predictive performance”

  •  

Response: ANN term was maintained.

  •  

Response: ANN term was maintained.

  •  

Response: Sentence shortened.

  •  

Response: TQWT was maintained.

  •  

Response: The novelity of determining the input data with the pTE approach in the study has been emphasized, and the following paragraph has been added.

In this study, a novel forecasting framework has been introduced for SPI data on 3- and 6-month time scales. The notation SPI-3 is used for the 3-month time scale, while SPI-6 is used for the 6-month time scale.

One of the unique contributions of this study is the utilization of a connectivity analysis using the pTE approach for SPI data prediction, specifically focusing on determining the optimal time lag values and SPI values at these time indices for input in the forecasting model. This novel approach represents the first-time in the literature for identifying the most suitable input SPI-3 and SPI-6 data with the optimal time indices for the prediction model. In this method, PTE values were calculated using a 1-3 phase delay from the Hankelization matrix obtained through time lag operations, thereby enabling the identification of time lags where information flow is at its maximum and minimum. By concentrating on time lags with maximum information flow, a methodology has been established for determining the most appropriate time indices and SPI inputs for use as input in SPI prediction.

Additionally, the TQWT method was employed to decompose SPI-3 and SPI-6 data into subbands, and the subband signals were subsequently estimated using ANN, SVR, and GPR models. The optimization of the TQWT method involved adjusting parameters like the Q factor, subband decomposition level, and the parameters of the machine learning models. This optimization was achieved through the GWO algorithm, resulting in the introduction of the GWO-TQWT-ML method, a novel approach in the literature known for its high forecasting performance with SPI data.  To evaluate the performance of the GWO-TQWT-ML method, we compared it against various models, including EMD-ANN, EMD-SVR, EMD-GPR, VMD-ANN, VMD-SVR, VMD-GPR, ANN, SVR, and GPR models, with the aim of benchmarking its performance. The study's results were meticulously assessed using performance metrics such as MSE, MAE, R, R2, and scatter plots. In summary, this study represents a significant advancement in SPI drought index estimation by introducing the novel pTE-GWO-TQWT-GPR approach. This approach leverages maximum information flow to determine input features for the forecasting model, and its incorporation significantly enhances forecasting accuracy.

  •  

Response: This title has been rewritten based on rewiever’s suggestions

In this study, precipitation data was utilized from the Catchment Attributes and Meteorology for Large-sample Studies (CAMELS) dataset [33]–[36]. The CAMELS dataset comprises observed runoff levels and catchment-aggregated meteorological forcing data at the daily timescale. Daily meteorological data were computed using gridded data sources, including the Daymet data [37], the Maurer data [38], and the National Land Data Assimilation System (NLDAS) data [39] in CAMELS dataset. Interested readers can find comprehensive information about this dataset [33]–[36].

For the proposed drought forecasting study, the Daymet data due to its high spatial resolution, is crucial for accurately estimating spatial variability in basins with complex topography. You can access these parameters freely at https://ral.ucar.edu/solutions/products/camels [33], [34]In this study, the SPI-3 and SPI-6 indices were computed using precipitation data collected from the gauge with ID 13340000 from the CAMELS dataset , spanning from October 1, 1980, to December 31, 2014. This gauge is situated in the Clearwater River basin in Orofino, characterized by a substantial drainage area of 14268.92 km2 and geographic coordinates at Latitude  46°28'42", Longitude 116°15'27". The study area is seen in Figure 2.

The precipitation data in this database are daily data and monthly averages of these data are taken as it is seen in Figure 3.

Figure 2. Study Area [40].

  •  

Response: location information restated in degrees (Latitude  46°28'42", Longitude 116°15'27)

  •  

Response: That sentence was rewritten.

  •  

Response: In the article, the sections specified by the referee were shortened and rewritten.

  •  

Response: The following explanation has been added to the article.

“In this study, we focused on the SPI-3, which represents precipitation conditions over a 3-month period, and SPI-6, which represents precipitation conditions over a 6-month period.”

  •  

Response: The following explanation has been added to the article.

In the analysis, 70% of the SPI data was allocated for training, while the remaining 30% was reserved for testing, without any randomization or reordering.

  •  

Response: The following explanation has been added to the article.

The novelity of this study lies in the application of the pTE approach, which is used for the first time in the literature to determine input variables and optimal time delay for forecasting purposes. pTE provides a robust and effective tool for identifying the op-timal time delay and input variables for the proposed drought forecasting model, al-lowing us to discern how one event influences another. Traditional methods often cause a time-consuming and complex analysis for forecasting. However, pTE stream-lines this process significantly while yielding more precise and efficient results. Thus, by using the pTE approach to determine the optimal time delays and input variables, it significantly contributes to improving forecasting performance and enables the de-velopment of prediction models in a more effective manner.

  •  

Response: The article was rewritten by making necessary summaries in this part of the article and other parts specified by the referee.

  •  

Response: The following explanation has been added to the article.

Phase transfer entrophy is a method that helps us gain insights by identifying rela-tionships between data. In this study, for the first time in the literature, a method using the pTE approach was proposed to predict SPI-3 and SPI-6 data based on their previ-ous values. The goal was to determine which past values of SPI data, known as "time lags", should be considered in the prediction process. This determination was made by analyzing the connection and information flow between the data at time (t-ς) and (t-τ)using the pTE values calculated from the Hankelization matrix of SPI data. As a result, the study aimed to enhance the performance of the prediction model by evalu-ating the calculated pTE values at different time indices and applying appropriately time-indexed data as inputs to the prediction model.

Additionally, a table has been added to show the performance of the lag time determination method with the pTE approach proposed in this study. Additions on this subject are listed below.

The Table 6 displays the prediction results obtained using inputs determined by pTE and inputs applied with consecutive time lags.

Table 6 The prediction performances of models where inputs corresponding to the recommended method's determined time indices are applied and models where inputs corresponding to consecutive time indices are applied.

Three Inputs - One Ahead Forecasting

SPI-6 (Three Inputs (t-1, t-2, t-3))

SPI-6 (Three Inputs (t-1, t-2, t-5), defined by proposed method)

Models

MSE

MAE

R

R2

MSE

MAE

R

R2

ANN

0.2957

0.4153   

0.8157   

0.6653

0.2800

0.4048

0.8271

0.6842

GPR

1.2777

0.6065

0.8129

0.6607

0.2832

0.4012

0.8241

0.6792

SVR

1.5763

1.0088

0.8177

0.6684

1.5623

1.0056

0.8245

0.6797

Three Inputs - Two Ahead Forecasting

SPI-6 (Three Inputs (t-2, t-3, t-4))

SPI-6 (Three Inputs (t-2, t-4, t-5), defined by proposed method)

Models

MSE

MAE

R

R2

MSE

MAE

R

R2

ANN

0.4983

0.5637

0.6677

0.4458

0.4844

0.5561

0.6761

0.4571

GPR

1.3651            

0.9464   

0.6689

0.4474

1.3508

0.9410

0.6692

0.4479

SVR

1.4105

0.9584

0.6660

0.4435

1.3793

0.9478

0.6713

0.4507

Three Inputs - Three Ahead Forecasting

SPI-3 (Three Inputs (t-3, t-4, t-5))

SPI-3 (Three Inputs (t-3, t-4, t-7), defined by proposed method)

Models

MSE

MAE

R

R2

MSE

MAE

R

R2

ANN

0.8767

0.7351

-0.025

0.0007

0.8520

0.7106

0.1013

0.0103

GPR

0.8855

0.7422

0.1041

0.0108

0.8490

0.7205

0.1345

0.0181

SVR

1.1638   

0.8726   

0.4998   

0.2498

1.1271

0.8570

0.5060

0.2560

For one-ahead forecasting of SPI-6 data, for example, SPI-6(t-1), SPI-6(t-2) and SPI-6(t-3) inputs are used as inputs to ANN, GPR, and SVR models. However, higher performance was obtained in the forecasting study using the SPI-6(t-1), SPI-6(t-2), and SPI-6(t-5) inputs recommended in this study for forecasting of SPI-6(t) data. According to these results, it can be observed that when inputs corresponding to the time indices determined by the proposed pTE-based approach are applied, the prediction performance is better than when inputs corresponding to consecutive time indices are applied.

  •  

Response: The following explanation has been added to the article.

The importance of SPI prediction lies in its ability to provide valuable insights into future precipitation patterns, aiding in water resource management, drought moni-toring, and climate adaptation efforts. In the literature, there are many methods that use machine learning (ML) approach for high performance prediction of SPI in recent years. A comprehensive review study on these methods can be found in the references. As observed in the literature [60], ML methods have proven to be highly effective in SPI prediction, yielding high-performance results. In this study, as a new approach compared to previous works, the optimal time lag value for SPI prediction using the past values of SPI data has been determined using phase Transfer Entrophy (pTE). Ad-ditionally, a novel pTE- Grey Wolf Optimization (GWO)- Tunable Q Wavelet Trans-form (TQWT)-ML method, utilizing preprocessed data with parameters optimized by GWO and the TQWT technique, has been proposed to enhance the performance of ML methods in SPI prediction

  •  

Response: It was fixed.

  •  

Response: The following explanation has been added to the article.

As seen in Table 6, when the method utilizing the recommended pTE approach with a specified time lag for SPI data is compared to the prediction results using consecutive time lag SPI data, it is evident that the prediction performance using input indices from the recommended method is superior.

  •  

Response: It was fixed.

  •  

Response: The following explanation has been added to the article.

TQWT and ML models, those parameters were optimized with GWO employed in developing forecasting models. In this process, SPI data underwent decomposition into sub-band signals using TQWT, and subsequently, these derived sub-band signals were predicted using ML algorithms, specifically Artificial Neural Network (ANN), Support Vector Machine (SVM) and Gaussian Process Regression (GPR).

  •  

Response: The following explanation has been added to the article.

The key results that address these contributions are as follows:

Introduction of a novel method incorporating time shifting processes and the phase transfer entropy approach for analyzing connectivity in time series data, mark-ing the first use of this approach in the field. Thus, identification of input data for SPI estimation using connectivity values derived from this method.

Utilization of TQWT subband signals in SPI estimation, which enhances the accu-racy of predictions.

Optimization of TQWT and ML model parameters through the GWO, improving the overall performance of the forecasting models.

Development of the pTE-GWO-TQWT-GPR model, which demonstrated the high-est forecasting performance for SPI data used in drought forecasting.

The potential applicability of the proposed approach not only for SPI data but also for estimating other time series related to hydrological processes and forecasting stud-ies.

These results collectively contribute to the advancement of methodologies for time series analysis and forecasting, particularly in the context of drought prediction and hydrological processes.

We believe that these revisions have significantly strengthened the manuscript, and we are confident that the revised version of the paper is now well-prepared for publication. We would like to express our gratitude for your time and effort in reviewing our work.

Thank you once again for your valuable feedback and guidance.

Sincerely,

Assoc. Prof. Dr. Levent LatifoÄŸlu

Erciyes University

Kayseri/Türkiye

Round 2

Reviewer 2 Report

I have read the response and the authors answered addressed my concerns